# The NCOR-HDAC3 co-repressive complex modulates the leukemogenic potential of the transcription factor ERG

Eitan Kugler[1,2,17], Shreyas Madiwale[1,3,17], Darren Yong[4,5], Julie A. I. Thoms [6,7], Yehudit Birger[1,3], David B. Sykes [8], Johannes Schmoellerl [9], Aneta Drakul[10], Valdemar Priebe[10], Muhammad Yassin[11], Nasma Aqaqe[11], Avigail Rein[1,3], Hila Fishman[1,3], Ifat Geron[1,3], Chun-Wei Chen [12,13,14], Brian Raught[4], Qiao Liu[12], Heather Ogana[15], Elisabeth Liedke [4,5], Jean-Pierre Bourquin[10], Johannes Zuber [9,16], Michael Milyavsky [11], John Pimanda [6,7], Gilbert G. Privé [4,5] ✉ & Shai Izraeli [1,3] ✉

The ERG (ETS-related gene) transcription factor is linked to various types of cancer, including leukemia. However, the specific ERG domains and co-factors contributing to leukemogenesis are poorly understood. Drug targeting a transcription factor such as ERG is challenging. Our study reveals the critical role of a conserved amino acid, proline, at position 199, located at the 3' end of the PNT (pointed) domain, in ERG's ability to induce leukemia. P199 is necessary for ERG to promote self-renewal, prevent myeloid differentiation in hematopoietic progenitor cells, and initiate leukemia in mouse models. Here we show that P199 facilitates ERG's interaction with the NCoR-HDAC3 co-repressor complex. Inhibiting HDAC3 reduces the growth of ERG-dependent leukemic and prostate cancer cells, indicating that the interaction between ERG and the NCoR-HDAC3 co-repressor complex is crucial for its oncogenic activity. Thus, targeting this interaction may offer a potential therapeutic intervention.

The human *ERG* gene belongs to the ETS family of transcription factors and is located on the long arm of chromosome 21 within the Down syndrome critical region[1–4]. In hematopoiesis, ERG supports self-renewal and functional properties of hematopoietic stem cells (HSC),

as observed in mice carrying an *Erg* germline mutation in the ETS DNA binding domain[5,6]. Similarly, *Erg* knockout in mice results in a rapid loss of self-renewing hematopoietic stem and progenitor cells (HSPC) due to premature differentiation[7]. Chromatin immunoprecipitation and

[1]Department of Human Molecular Genetics and Biochemistry, Faculty of Medicine, Tel Aviv University, Tel Aviv, Israel. [2]Institute of Hematology, Davidoff Cancer Center, Rabin Medical Center, Petah Tikva, Israel. [3]The Rina Zaizov Pediatric Hematology and Oncology Division Schneider Children's Medical Center of Israel, Petach Tikva, Israel. [4]Princess Margaret Cancer Centre, University Health Network, Toronto, ON, Canada. [5]Department of Biochemistry, University of Toronto, Toronto, ON, Canada. [6]Adult Cancer Program, Lowy Cancer Research Centre, UNSW Sydney, Sydney, NSW, Australia. [7]School of Biomedical Sciences, Faculty of Medicine, UNSW Sydney, Sydney, NSW, Australia. [8]Center for Regenerative Medicine, Massachusetts General Hospital, Boston, MA, USA & Harvard Stem Cell Institute, Cambridge, MA, USA. [9]Research Institute of Molecular Pathology (IMP), Vienna BioCenter (VBC), Vienna, Austria. [10]Division of Pediatric Oncology, and Children Research Center, University Children's Hospital, Zurich, Switzerland. [11]Department of Pathology, Sackler Faculty of Medicine, Tel Aviv University, Tel Aviv, Israel. [12]Department of Systems Biology, Beckman Research Institute, City of Hope, Duarte, CA, USA. [13]Department of Pediatric Oncology, Dana-Farber Cancer Institute, Harvard Medical School, Boston, MA, USA. [14]City of Hope Comprehensive Cancer Center, Duarte, CA, USA. [15]Department of Pediatrics, Division of Hematology and Oncology, Children's Hospital Los Angeles, University of Southern California, Los Angeles, CA, USA. [16]Medical University of Vienna, Vienna, Austria. [17]These authors contributed equally: Eitan Kugler, Shreyas Madiwale. ✉e-mail: Gil.Prive@uhnresearch.ca; sizraeli@gmail.com

sequencing (ChIP-Seq), integrated with gene expression analysis demonstrated that ERG is part of a "heptad" of transcription factors (TAL1, LYl1, LMO2, GATA2, RUNX1, ERG, FLI1), physically and functionally interacting with key cis-regulatory regions in HSC. Combinatorial interactions between these transcription factors direct HSPC function and fate[8–10].

When aberrantly expressed, ERG is a hematopoietic oncogene. High levels of ERG expression were associated with decreased overall survival in a study of 84 patients with normal karyotype acute myeloid leukemia (AML)[11]. In another cohort of 210 AML patients with normal karyotype, ERG expression was one of the strongest predictors of treatment outcome in a multivariable analysis[12]. We previously reported a mouse transgenic model for human ERG designed to clarify if ERG directly promotes acute leukemia[13,14]. By 5 months of age, mice developed either T-ALL (30% of cases) or AML (70% of cases), characterized by a stem cell gene expression signature similar to human AML[13]. However, the molecular mechanisms that account for ERG oncogenesis in acute leukemia are mostly unknown.

Chromosomal translocations and structural aberrations related to ERG have been reported in AML, including the *FUS-ERG* and the *ELF4-ERG* fusion genes[15,16]. Moreover, a RAG-dependent *ERG* deletion or alternative transcript, following DUX4-IGH rearrangement, has been shown to promote B-cell precursor acute lymphoblastic leukemia (ALL) with a favorable outcome[17,18]. Co-expression of ERG and the short isoform of Gata1 (Gata1s, universally expressed in Down syndrome's myeloid malignancies), led to transient myeloproliferative disorder and acute megakaryocytic leukemia (AMKL) in mice[19–23]. Furthermore, ERG is associated with oncogenic protein complexes in human leukemia. For instance, the *CBFA2T3 (ETO2, MTG16)-GLIS2* fusion gene hijacks ERG to modify gene expression and the chromatin landscape of megakaryocytic precursors, resulting in differentiation arrest and AMKL development[24]. In another AML subtype associated with ETO proteins, ERG co-binds genomic regions with RUNX1-RUNX1T1 (ETO, MTG8, CBFA2T1) and its interacting partners, RUNX1, P300, and HDACs. The activity of ERG modulates RUNX1-RUNX1T1 expression, prevents oncogene overexpression, and maintains leukemia viability[25]. Finally, a recent study has demonstrated that ERG, in conjunction with members of the SWI-SNF complex and P300, mediates the leukemogenic process of the *TCF3-HLF* fusion gene in B lymphoid progenitors[26].

We demonstrated in a transgenic mouse that ERG is a leukemia oncogene[13,20]. Here, we report that P199, a conserved proline at the C-terminal end of the ERG PNT domain, is essential for leukemic transformation. We further demonstrate that ERG interacts with the NCoR-HDAC3 complex in a P199-dependent manner to maintain HSPCs in an undifferentiated state. Consistent with this result, the chemical and genetic inhibition of HDAC3 in ERG-dependent human leukemia cells suppresses cell growth in vitro and restricts leukemia progression in vivo.

## Results

### Mutation of a single amino acid in ERG abrogates leukemia development

To investigate the molecular mechanisms implicated in ERG-driven AML, we designed several transgenic mouse lines in which the expression of human ERG is regulated by the VAV promoter in HSPCs (TgERG). All mice developed AML or T-ALL by 5 months of age[13,14,20]. Fortuitously, during the construction of the first line of the ERG transgenic mice, a PCR error resulted in the substitution of an evolutionary conserved proline at position 199 by leucine (Fig. 1A and Fig. S1A, S1B). The mutant *ERG* RNA and protein were expressed in the spleen of transgenic mice (Fig. S1C). Immunoblot analysis of transfected CMK cells demonstrated that P199L ERG retains the nuclear localization of ERG (Fig. 1B). To study the effect of P199L on the structure and thermostability of the protein, we generated truncated and full-length versions of human ERG in *E. coli* (Fig. S2A). High-performance liquid (HPLC) and size-exclusion chromatography (SEC) were used to test whether P199L affects the hydrodynamic properties of ERG. Purified WT and P199L-ERG are monomeric in solution and monodisperse in SEC. The elution volumes of WT and P199L-ERG were indistinguishable, indicating that the mutation does not affect the hydrodynamic properties of the protein (Fig. 1C). Circular dichroism spectroscopy was used to assess differences in conformation and stability between ERG variants. The secondary structure and the thermostability of ERG were not significantly affected by P199L (Figs. 1D, E and S2B, C). Strikingly, transgenic mice harboring the P199L variant were viable and did not show any signs of leukemia during their entire lifespan (Fig. S1D). Therefore, the P199 conserved amino acid plays a crucial role in ERG-mediated leukemogenesis. Regrettably, this

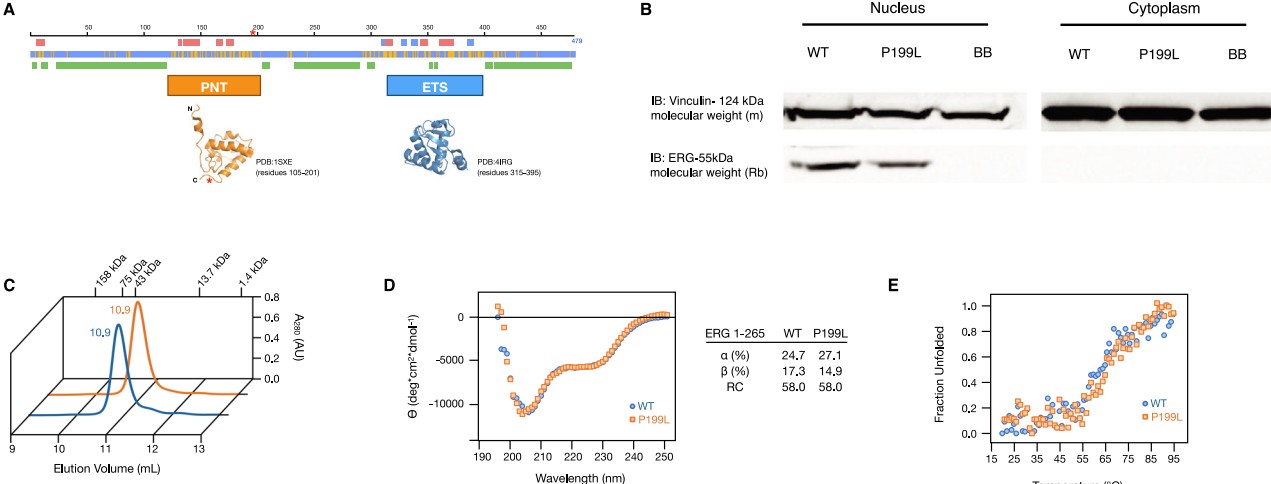

**Fig. 1 | P199L does not significantly affect the secondary structure or thermal stability of ERG. A** Sequence-based prediction of ERG, including secondary structure (red: helix, blue: beta-strand), solvent exposure (blue: exposed, yellow: buried), and disorder (green) (http://www.predictprotein.org). Experimentally determined structures of the ordered domains are also shown. Proline 199 is at the end of the PNT domain. **B** Immunoblot for FLAG in the nuclear and cytoplasmic fractions of CMK cells transduced with ERG variants. **C** Analytical size-exclusion chromatogram of the ERG constructs. Protein size standards are indicated. **D** Circular dichroism spectrum scan at 20 °C post-melt. **E** Circular dichroism spectrum collected at 226 nm in a temperature gradient. WT−Wild-type ERG, BB−Backbone. Source data are provided as a Source Data file.

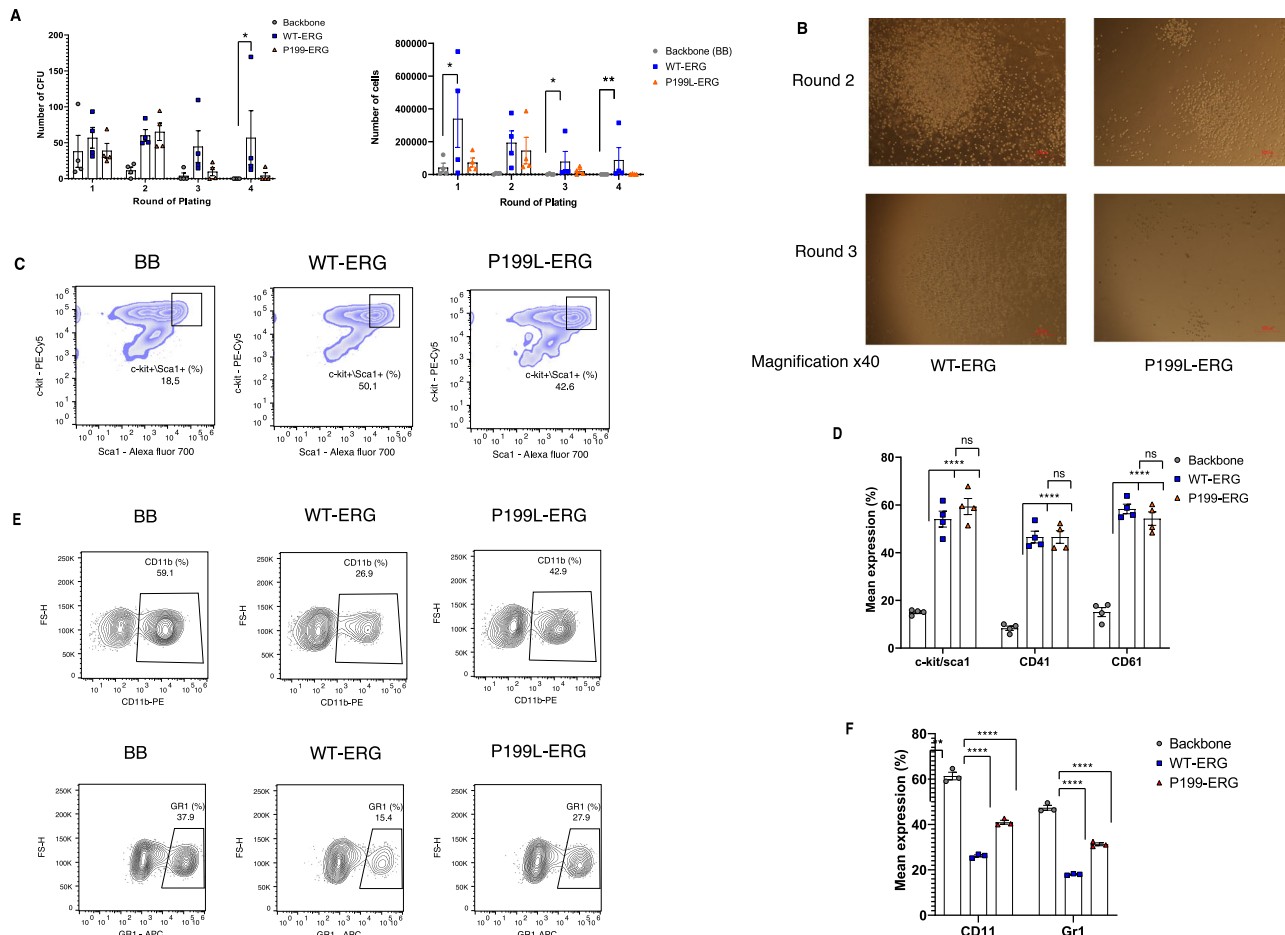

**Fig. 2 | P199L alleviates the effect of ERG forced expression on self-renewal and myeloid differentiation of HSPC. A** Re-plating assay in semi-solid conditions of murine fetal liver-derived HSPC transduced with ERG variants. The number of CFU (left panel) and total number of cells collected (right panel) per $10^4$ cells plated is presented for each round of plating. Data is represented as mean ± SEM, two-way ANOVA, and Tukey's multiple comparison test. The *P*-value for CFUgraph comparing BB vs. WT-ERG on round plating is 0.0398. *n* = 4 independent experiments in each experiment for each biological condition *n* = 1 biological independent sample. **B** Microscopic view of colonies morphology between groups. MagnificationX40. **C** Representative flow cytometry analysis of murine HSPC (c-Kit, Sca1) cell surface markers. **D** Summary bar graphs of flow cytometry analysis for the end of round one. *n* = 4 independent experiments in each experiment for each biological condition *n* = 1 biological independent sample. Data is represented as mean with SD, *t*-test, Mann–Whitney, unpaired. The *P*-value BB vs. WT-ERG/P199L for cKit/Sca1, CD41, and CD61 is <0.0001. **E** Representative flow cytometry analysis of myeloid differentiation using measurements for cell surface expression of CD11b and Gr-1 in fetal-liver-derived HSPC transduced with ERG variants. **F** Summary bar graphs of flow cytometry analysis right panel, *n* = 3 independent experiments in each experiment for each biological condition *n* = 1 biological independent sample. Data is represented as mean ± SEM, two-way ANOVA, and Tukey's multiple comparison test. The *P*-value BB vs. WT-ERG/P199L for CD11b and Gr1 is <0.0001. CFU−Colony-forming units, * denotes *P* ≤ 0.05, ** denotes *P* ≤ 0.01, *** denotes *P* ≤ 0.001, **** denotes *P* ≤ 0.0001. Source data are provided as a Source Data file.

first transgenic line of mice was discarded, thus no further analysis is possible.

## P199L decreases the leukemic transformation of HSPC induced by ERG expression

As P199L does not compromise the 3D structure, thermostability, or nuclear localization of ERG, we sought to evaluate its functional effects on HSPCs in vitro and in vivo. ERG dosage has been found to be critical to endow HSPC with aberrantly high self-renewal potential[7,19,27]. We, therefore, hypothesized that the absence of leukemia in P199L-ERG transgenic mice was due to the decrease in self-renewal capacity of the transgenic HSPCs. To test this hypothesis, murine fetal liver-derived HSPC (C57BL/6 mice, E13.5, c-Kit⁺/lineage negative cells) were transduced with a lentivirus expression vector of human ERG variants, followed by a re-plating assay in semi-solid conditions. HSPC transduced with P199L-ERG exhibited impaired self-renewal and colony-forming capacity as compared with the WT-ERG (Fig. 2A, B). The reduction in colony formation was not accompanied by a significant increase in apoptosis (Fig. S3).

Next, we examined whether P199L-ERG expression influenced the lineage commitment of HSPC. We used flow cytometry to measure the cell surface expression of murine HSPC markers c-Kit and Sca1, as well as megakaryocytic and myeloid cell markers, 72 h after viral transduction with ERG variants (Fig. 2C–F). As we have previously shown[13,22] ERG increased the percentage of c-Kit/Sca1 double positive HSPCs with megakaryocytic features, while blocking myeloid maturation. The P199L variant had the same effect as the WT-ERG on HSPCs and megakaryocytic markers but failed to block myeloid differentiation (Fig. 2C−F, expression of CD11b and Gr-1, *P* < 0.05). We further evaluated the functional effects of P199L-ERG on the megakaryocytic lineage by transducing both WT and mutant ERG into K562 cells (erythroleukemia with low expression of endogenous ERG). Similar to what was previously demonstrated for WT-ERG[19,22], P199L-ERG induced an erythroid to megakaryocytic switch (Fig. S4A). This result was also confirmed by a CD41 promoter luciferase reporter assay in 293T cells (Fig. S4B).

To independently confirm our initial observation in the transgenic mice regarding the leukemogenic potential of the WT and P199L-ERG isoforms, we carried out a series of transduction-transplantation

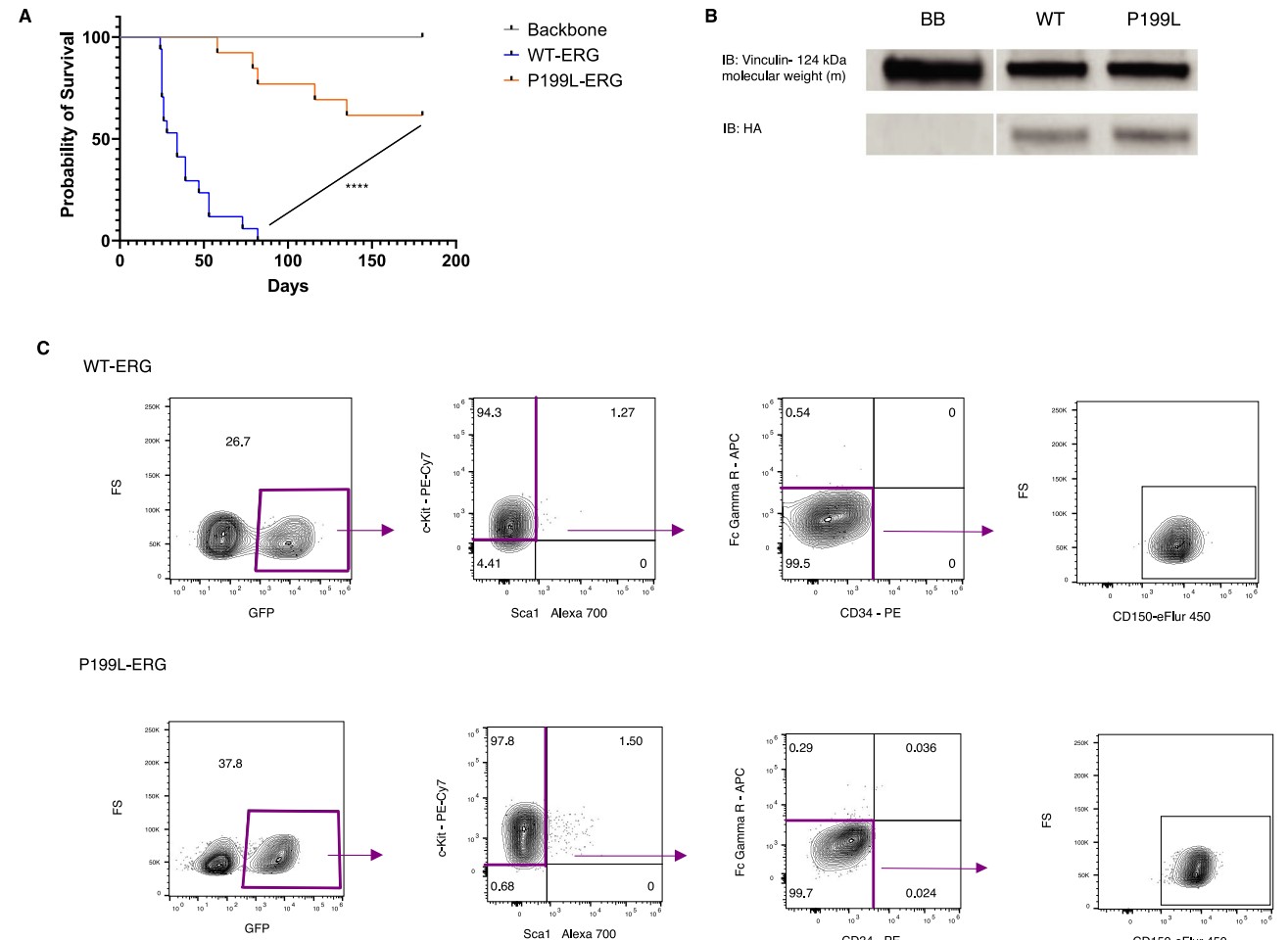

**Fig. 3 | P199L severely disrupts ERG transforming capacity in vivo. A** Survival analysis of C57B/6 mice transplanted with murine fetal liver-derived HSPC following transduction with ERG variants ($10^5$ transduced cells/mouse). A log-rank test was used to compare survival distribution between groups. The *P*-value comparing WT-ERG vs. P199LERG is *P* < 0.0001. * denotes $P \le 0.05$, ** denotes $P \le 0.01$, *** denotes $P \le 0.001$, **** denotes $P \le 0.0001$. **B** Immunoblot of HA-tagged ERG to compare the levels of protein expression of WT and mutated ERG in the spleen of leukemic mice. **C** Immunophenotype of AML blasts (GFP+) from the bone marrow of leukemic mice. Source data are provided as a Source Data file.

experiments. Murine fetal liver-derived HSPCs transduced with human ERG variants were transplanted into sub-lethally irradiated C57BL/6 mice. During the 180-day observation period, all WT-ERG mice developed AML compared with 37.5% of P199L-ERG mice. Furthermore, the latency period for leukemia development was significantly longer in mutant mice (Fig. 3A, B) (Log-rank test, *P* < 0.0001). The immunophenotype was comparable to what we previously reported for TgERG leukemia regardless of ERG variant, namely lineage negative, c-Kit^low, CD150$^+$ AML. This population of cells is enriched with mega-erythroid progenitors (MEP, Fig. 3C)[13,28].

As P199 is the last amino acid in the PNT domain, we hypothesized that the P199L-ERG phenotype could result from the loss-of-function of the entire domain. To test this hypothesis, murine fetal liver-derived HSPC were transduced with an ERG construct lacking the PNT domain (delPNT-ERG), and transduction-transplantation experiments were performed as described above for P199L-ERG. Murine HSPC expressing delPNT-ERG retains self-renewal ability in re-plating assays and colony-forming unit assays and progresses to AML upon transplantation to immune-deficient mice (Fig. S5). Furthermore, the morphology of delPNT-ERG colonies in semi-solid conditions differs from that of WT and P199L-ERG colonies. Colonies were more widely dispersed and tended to be more proliferative (Fig. S5A).

Taken together, ERG-induced leukemic transformation is coupled with enhanced self-renewal capacity and restriction of myeloid differentiation in HSPCs. P199L disrupts these oncogenic functions but does not result in a complete loss of function as other features related to ERG expression in HSPC as megakaryopoiesis were not affected. The phenotype of P199L-ERG was not equivalent to the phenotype of delPNT-ERG.

Next, we tested if P199L-ERG expression in HSPC has a dominant negative effect on endogenous murine Erg. Endogenous Erg in murine fetal liver-derived HSPC has no direct role in leukemia progression since HSPC transduced with an empty vector and transplanted to immunocompetent mice do not develop leukemia (Fig. 3A). However, a dominant negative effect of P199L-ERG may affect self-renewal and myeloid differentiation. Since silencing endogenous *Erg* in primary murine fetal liver progenitor cells is lethal, as previously reported[7,29], complementary assays were required to determine whether P199L-ERG had a dominant negative effect over endogenous ERG. To this end, we utilized an ERG + 85 stem cell enhancer luciferase reporter assay in 293 T cells. This enhancer contains binding motifs for multiple transcription factors, including ERG itself (heptad of transcription factors−SCL/TAL1, LMO2, LYL1, ERG, FLI1, RUNX1, GATA2), and its activation is highly correlated to the expression of ERG in human AML samples[30]. We hypothesized, therefore, that a dominant negative effect of P199L-ERG may negatively affect the transactivation of the *ERG + 85* enhancer and disrupt endogenous *Erg* expression in murine HSPCs. Forced expression of P199L-ERG in 293 T cells reduced

transactivation of the enhancer compared to WT-ERG (Fig. S6A). However, the co-expression of WT and P199L-ERG activated the enhancer similarly to the expression of WT-ERG alone suggesting against a negative dominant effect of P199L-ERG (Fig. S6B).

Additionally, we studied the possible dominant negative effect of P199L-ERG in ERG-dependent human AML cells. P199L-ERG was expressed in SKNO1 cells (RUNX1-RUNX1T1 rearrangement) stably expressing a lentiviral reporter system in which BFP expression is driven by *ERG* + 85 as we previously described (a decrease in transactivation of the enhancer would result in a decrease in BFP expression)[30]. No significant decrease in BFP signal nor in cell proliferation was observed after expression of P199L-ERG, suggesting that mutated ERG has no dominant negative effect over endogenous ERG in SKNO1 cells (Fig. S7).

## P199L compromises the ERG transcriptional repression signature in HSPC

To investigate how P199L affects ERG's transcriptional signature, RNA sequencing (RNA-seq) of murine fetal liver-derived HSPCs expressing ERG variants was performed. Cells were cultured in serum-free media supplemented with SCF, TPO, and FLT3, and collected for gene expression analysis 72 h after transduction. Differentially expressed gene analysis indicated that P199L interferes with ERG-mediated transcriptional repression. As shown in Fig. 4A, gene sets that correlate with upregulated genes are relatively evenly distributed among ERG variants. By contrast, transcriptional signatures associated with downregulated genes are highly enriched for WT-ERG. These sets of genes include those found to be repressed in HSPC transformed by other leukemia-related oncogenes (HOXA9-MEIS1, NUP98-HOXA9). Interestingly, genes that were upregulated upon NUP98-HOXA9 overexpression in HSPC ("NUP98-HOXA9 OE in HPC−upregulated genes") were evenly enriched for both ERG variants. As previously demonstrated, *ERG* has been identified as a direct target of MEIS1, together with other stemness-associated genes, such as *CD34, FLT3,* and *MYC*[31–34]. Our analysis reveals that the ectopic expression of ERG represses a common set of target genes during AML development, which is compromised by the P199L mutation.

Moreover, specific genes normally upregulated during myeloid differentiation were transcriptionally downregulated in WT-ERG-expressing cells compared with cells expressing P199L-ERG ("Myeloid differentiation−upregulated genes", Fig. 4A−C), supporting the observed immunophenotype (Fig. 2E, F). Furthermore, CBFA2T3 transcriptional repression signatures in HSPC strongly correlated with WT-ERG (Fig. 4A−C). The transcriptional co-repressor factor CBFA2T3 recruits the NCoR/SMRT complex together with class I HDACs to regulate HSC fate[35,36]. In addition, it plays a crucial role in the development of pediatric AMKL in which CBFA2T3-GLIS2 interacts with ERG to promote leukemogenic activity[24]. Taken together, these results indicate that P199L substitution negatively affects ERG-mediated repression of defined target genes, which are essential for AML development.

WT and mutated human ERG samples display no significant difference in endogenous *Erg* expression, as discussed in the previous section (Fig. S8, left panel.)

## P199L disrupts the interaction of ERG with chromatin modifiers

P199 is the last amino acid of the PNT domain, which is structurally similar to the SAM protein interaction domain[37–39]. We hypothesized that P199L might, therefore, affect the interaction of ERG with one or more proteins essential to the progression of leukemia. We used proximity ligation-mass spectrometry (BioID) to compare proteins that localize in the proximity of WT vs. P199L ERG in HEK293 cells[40]. A total of 240 putative protein interactors with a significance analysis of interactome (SAINT)[41] scores >0.8 and false-discovery rate (FDR) < 0.05 were identified. The hits were highly enriched with chromatin

modifiers (Fig. S9), including 25 proteins associated with histone deacetylation. Notably, all the core components of the NCoR-HDAC3 complex were identified, including HDAC3, NCoR1, NCoR2, TBL1XR1, and GPS2, increasing the likelihood that the interaction is functionally meaningful (Fig. S9B). The second most prominent complex identified is the SWI/SNF (BAF) chromatin remodeling complex. We have identified 10 of the 14 core components of the BAF complex: SMARCB1, SMARCC1, SMARCC2, SMARCD1, SMARCD2, SMARCE1, ACTL6A, SMARCA4, ARID1A and ARID1B (Fig. S9B). In support of this result, it was recently shown by the SILAC-based approach followed by mass spectrometry, that the TMPRSS2-ERG fusion oncoprotein interacts with canonical members of the SWI/SNF complex in a prostate cancer cell line[42].

To assess the effect of the P199L mutation on the ERG BioID hits, we compared the proximity maps of WT-ERG and P199L-ERG (Fig. 5A, B and Table S1). Globally, the P199L mutation resulted in a lower number of spectral counts in many of the hits, suggesting an impaired interactome. Remarkably, the largest effects were seen with the members of the NCoR-HDAC3 complex, with an approximate 40% reduction in the number of spectral counts for NCoR-HDAC3 complex members in comparison with WT-ERG (Fig. 5A, B).

To verify whether disruption of NCoR-HDAC3 complex interaction with ERG by P199L affects chromatin modifications of genes associated with leukemogenesis we stably expressed human WT and P199L ERG in the ER-Hoxb8 cell model of conditionally transformed hematopoietic progenitors that exhibit myeloid differentiation potential upon beta-estradiol withdrawal[43]. We used these isogenic cell lines to systematically investigate the global effect of ectopic ERG and P199L-ERG expression on chromatin modifications by conducting chromatin immunoprecipitation (ChIP) sequencing coupled with transcriptome analysis.

We observed subtle differences in the global statistics of the ChIP signal enrichments (Fig. 6A). As we suspected an acetylation/deacetylation perturbation with P199L-ERG due to the decreased interaction with NCoR-HDAC3 complex, we focused the analysis on the distribution of H3K27ac across the samples. We measured acetylation intensity at gene loci that exhibited differential expression and repression upon WT-ERG forced expression. We observed a decrease in the H3K27ac signature at over 1500 unique sites in cells transduced with WT-ERG compared to those transduced with the mutant (Fig. 6B, C). These gene loci were relatively hyperacetylated in both the control and P199L-ERG cells, compared to ER-Hoxb8 cells expressing WT-ERG (Fig. 6B, C). We observed less variation in the intensity of other types of histone modifications at those specific loci (Fig. 6B). These sites were predominantly associated with enhancers (co-occupancy of H3K27ac and H3K4me1) of genes expressed in mature myeloid cells (Fig. 6D, E). Intriguingly, variations in H3K27ac intensity between groups were less pronounced at transcription start sites (co-occupancy of H3K27ac, H3K4me3, and H3K9ac) (Fig. S10).

Collectively, these findings suggest that overexpression of WT-ERG in myeloid progenitors reduces chromatin acetylation and activity at enhancer regions of genes expressed in mature myeloid cells. This in turn may restrict myeloid differentiation, as consistently demonstrated in AML.

## HDAC3 inhibition alleviates the myeloid differentiation block induced by ERG expression in hematopoietic progenitors

As demonstrated, the expression of ERG in HSPCs restricted myeloid differentiation (Fig. 2A) and was accompanied by repression of myeloid differentiation genes via a decrease in the H3K27ac levels (Figs. 4 and 6). The proximity protein interaction map of ERG coupled with ChIP-sequencing data suggested that this repression may be mediated by the recruitment of NCoR-HDAC3 repression complex to these ERG trans repression targets. To examine this hypothesis, we used the ER-Hoxb8 cells stably expressing human ERG as a model of myeloid

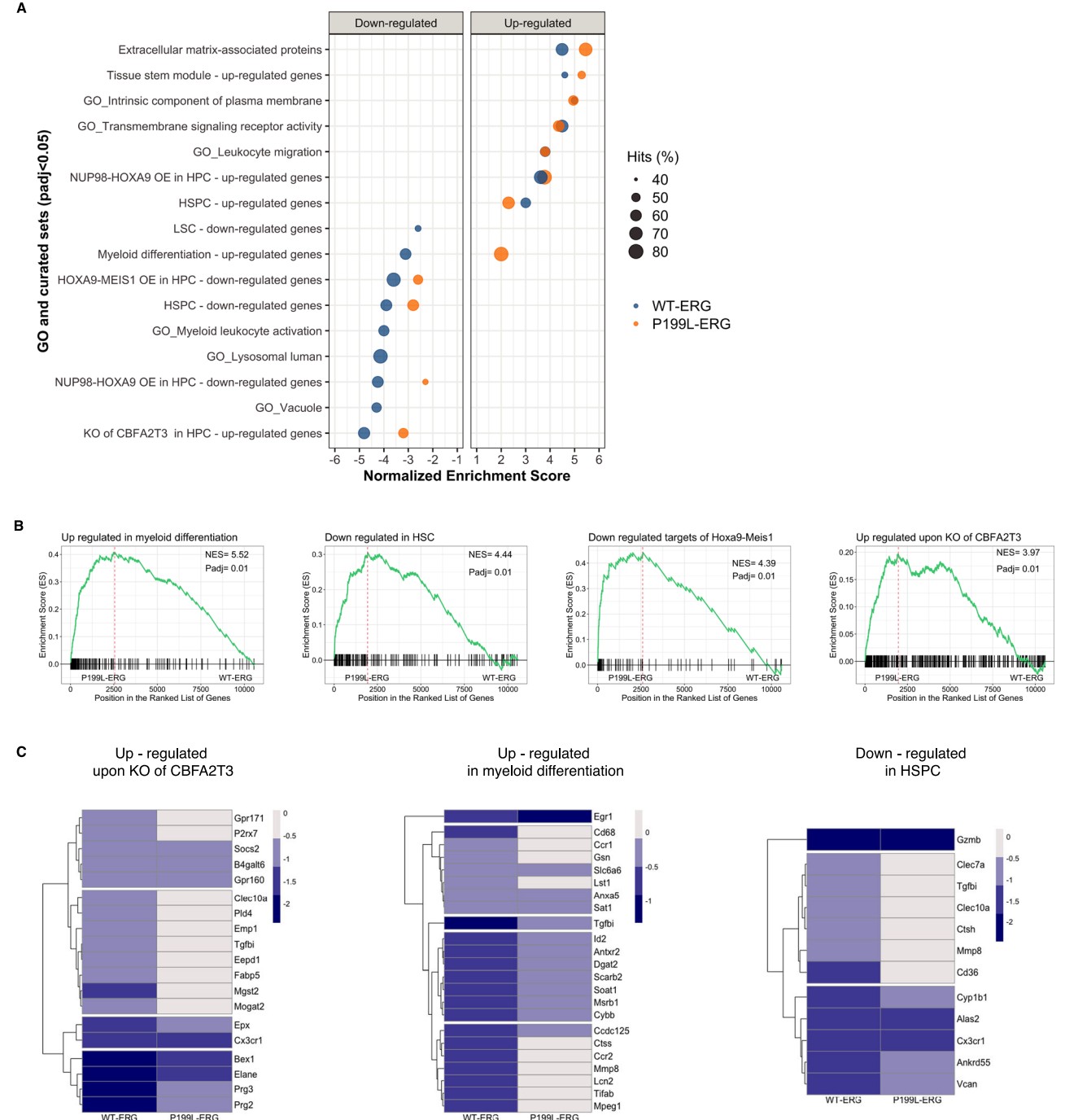

**Fig. 4 | P199L impedes ERG-mediated transcriptional gene repression in HSPC.**
**A** Pathway and functional analysis of the gene expression profile of ERG variants against BB was performed using GSEA. Molecular pathways related to down and upregulated genes are presented separately. The absence of a circle implies no statistical significance enrichment for a particular gene set. A positive enrichment score refers to upregulated genes, and a negative score refers to downregulated genes (ERG variants against the backbone). The percentage of hits ("tags" in leading-edge analysis) is displayed to indicate the percentage of genes contributing to the enrichment score for each set of genes. **B** GSEA of upregulated genes (P199L-ERG against WT-ERG). **C** Most significant DEG ($P_{adj} < 0.05$) contributing to the enrichment scores in A are shown for representative sets (ERG variants against BB). HSPC—hematopoietic stem and progenitor cells, LSC—leukemic stem cells. NES—Normalized enrichment score. DEG—differentially expressed genes. BB—backbone. Source data are provided as a Source Data file.

differentiation. Upon removal of beta-estradiol from the growth medium, the ER-Hoxb8 cells differentiate into mature granulocytes over the course of 5 days[43] (Fig. 7A).

As was demonstrated for murine fetal liver-derived HSPCs (Fig. 2E, F), the proportion of Gr1-positive cells (marker for mature granulocytes) was significantly lower for ERG-expressing ER-Hoxb8 cells in comparison to the cells transduced with backbone (BB) control vector (Fig. 7A). Consistent with the interaction of ERG with the NCoR-HDAC3 repressor complex, specific HDAC3 inhibitors (BRD3308 and RGFP966)[44,45] alleviated the block in myeloid differentiation induced by ERG in ER-Hoxb8 cells (Fig. 7B). Moreover, the self-renewal capacity of murine fetal liver-derived HSPC transduced with ERG was also compromised upon HDAC3 inhibition (Fig. S11).

Next, we addressed the role of HDAC3 in ERG's transcriptional activity. We treated human ERG-expressing ER-Hoxb8 cells either with BRD3308 or DMSO for 48 h and performed RNA sequencing. Similar to

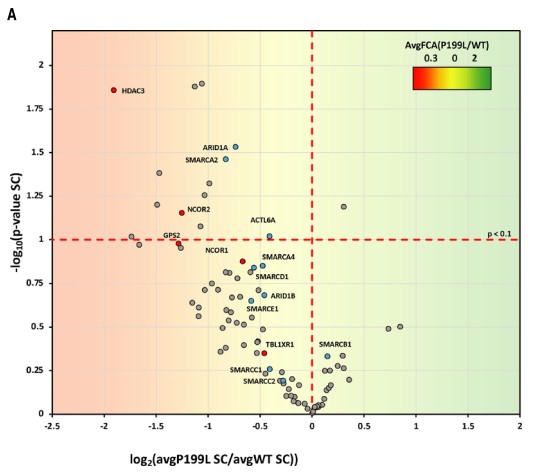

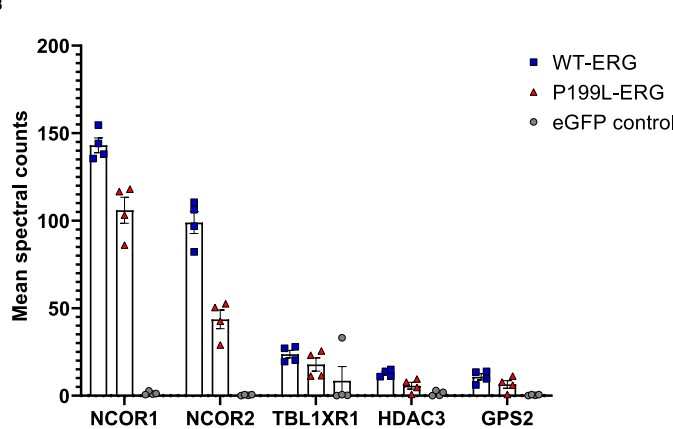

**Fig. 5 | P199L disrupts the interaction of ERG with chromatin modifiers.**
**A** Volcano plot comparing the spectral counts of identified interactors between FLAG-BirA-ERG WT and P199L. The background color indicates the prey FCA P199L/WT ratio. Significant proteins were selected with a $P < 0.08$ cutoff based on spectral count ratios. Notable proteins are colored according to molecular function. **B** Mean spectral counts for NCOR-HDAC3 hits (averaged between two biological replicates). Saint scores and bFDR are >0.8 and <0.05, respectively. FCA−fold-change. SC−spectral counts. Source data are provided as a Source Data file.

murine fetal liver-derived HSPC cells, WT and mutated human ERG samples did not demonstrate significant differences in endogenous *Erg* expression. Endogenous *Erg* expression was lower in cells trans-duced with an empty vector (BB). This is likely due to the more mature differentiation state of these cells as compared to those that express ERG variants (Fig. S8B). It was previously demonstrated in ER-Hoxb8 cells that the expression of *Erg* as well as other stem cell transcription factors such as *Gata2* decreases as the cells differentiate upon removal of estrogen from the culture medium (Table S2).

As shown in Fig. 7C, HDAC3 inhibition was associated with both de-repression and repression of target genes, but the effect on de-repression was more pronounced. Interestingly, leukemia-associated pathways that were regulated by ERG in murine fetal liver-derived HSPCs (Hoxa9-Meis1, CBFA2T3, and repression of myeloid differ-entiation genes) were ranked at the leading edge of gene sets that were significantly perturbed after HDAC3 inhibition (Fig. 7D). Particularly sensitive to HDAC3 inhibition were canonical pathways associated with differentiated myeloid cells (Fig. 7E, F). Figure 7G illustrates the most significant myeloid-related genes repressed by ERG expression in Hoxb8 cells. These genes are also repressed by P199L-ERG, although to a lesser extent, supporting the hypothesis that P199L interferes with myeloid restriction. Moreover, there is a subtle difference in the magnitude of gene repression between WT and mutated ERG, sug-gesting that P199L is not a complete loss of function mutation. Inhi-biting HDAC3 results in the de-repression of these genes.

Taken together, the gene expression pattern and the cell pheno-type (Fig. 2F) are in agreement with a decreased association of the NCoR-HDAC3 repression complex associated with P199L-ERG rather than a complete disruption of the interaction. This subtle effect of P199L on ERG is sufficient to significantly inhibit leukemogenesis, as discussed earlier.

**HDAC3 inhibition attenuates ERG-dependent human leukemias and prostate cancer**
The Bio-ID proximity assay was performed on 293T cells following transfection with ERG variants. To validate the ERG/NCoR-HDAC3 interaction in models of established leukemia, we confirmed the co-immunoprecipitation of the NCoR-HDAC3 complex with endogenous ERG in human AML cells (Fig. 8A). Next, we treated human AML cells expressing high and low levels of ERG (Fig. S12)[30] with an HDAC3 inhibitor (RGFP966) and examined its effect on cell proliferation. We reassessed the ERG dependency of these leukemic lines by Cas9-mediated targeting of ERG. We found that genome editing of ERG

resulted in the loss of leukemia cell growth accompanied by elevated apoptosis in the case of SKNO1 (AML harboring the RUNX1-RUNX1T1 translocation) and TF1 (erythroleukemia) cells, while THP1 cells were not affected by the ERG inactivation (Fig. 8B, C and Fig. S12B). In agreement with their ERG dependency, SKNO1, and TF1 cells were more sensitive to HDAC3 inhibition (Fig. 9A and Fig. S12C).

Schmoellerl et al. reported recently that EVI1-driven poor prog-nosis AMLs are ERG-dependent[46]. Consistent with their discovery, HNT-34, EVI1-driven AML cells were highly sensitive to HDAC3 inhibi-tion (0.1 micromolar RGFP966 was sufficient to significantly decrease cell growth) as compared to EVI1/ERG-independent MLL (KMT2A)-rearranged AML cells (Fig. 9B).

To examine whether the immunophenotype of ERG-dependent AML cells alters upon HDAC3 inhibition, flow cytometry analysis was conducted 96 h following treatment with RGFP966 on SKNO1, TF1, and ELF-153. Both TF1 (erythroleukemia, CD3-, CD13+, CD14-, CD15-, CD19-, CD33+, CD34+, CD71+, HLA-DR+) and ELF-153 (hypodiploid AML, CD3-, CD4+, CD13+, CD14-, CD15-, CD19-, CD33+, CD34+, HLA-DR+) displayed features of monocytic differentiation as demonstrated by a significant increase in CD14, CD36, CD64, and HLA-DR MFIs (Fig. S13).

ERG dependency is not limited to AML. According to the DepMap illustrated in Fig. S14A, numerous ALL cell lines are dependent on ERG. Specifically, in the poor prognosis B-ALL type harboring the TCF3-HLF translocation, ERG cooperates with TCF3-HLF to regulate enhancer functions and is essential for leukemia maintenance[26]. Moreover, a drug screen conducted previously on the HAL01 cell line (TCF3-HLF fusion) demonstrated high sensitivity to RGFP966 (Fig. S14B)[47]. We have therefore examined the effect of HDAC3 inhibition on the survival of patient-derived xenograft (PDX) of TCF3-HLF B-ALL as was pre-viously described[48]. As shown in Fig. S14C, the PDX samples were sensitive to RGFP966 at concentrations comparable to those used for AML cell lines.

In addition, more than 50% of cases of prostate cancer are asso-ciated with the TMPRSS2-ERG fusion gene rearrangement[49]. We there-fore investigated the effect of HDAC3 inhibition on the survival of prostate cancer cells. The ERG-NCoR-HDAC3 interaction was first con-firmed in VCaP cells by Co-IP (Fig. S15A). As demonstrated in Fig. S15B, VCaP cells (TMPRSS2-ERG-driven) exhibited higher sensitivity to HDAC3 inhibition than LNCaP cells (low levels of endogenous ERG expression). Interestingly, the expression of NCoR2-HDAC3 proteins in LNCaP cells, that are independent of ERG was relatively low (Fig. S15C).

To evaluate a possible anti-leukemic effect of HDAC3 inhibition in vivo we utilized the SKNO1 leukemia model bearing a RUNX1-

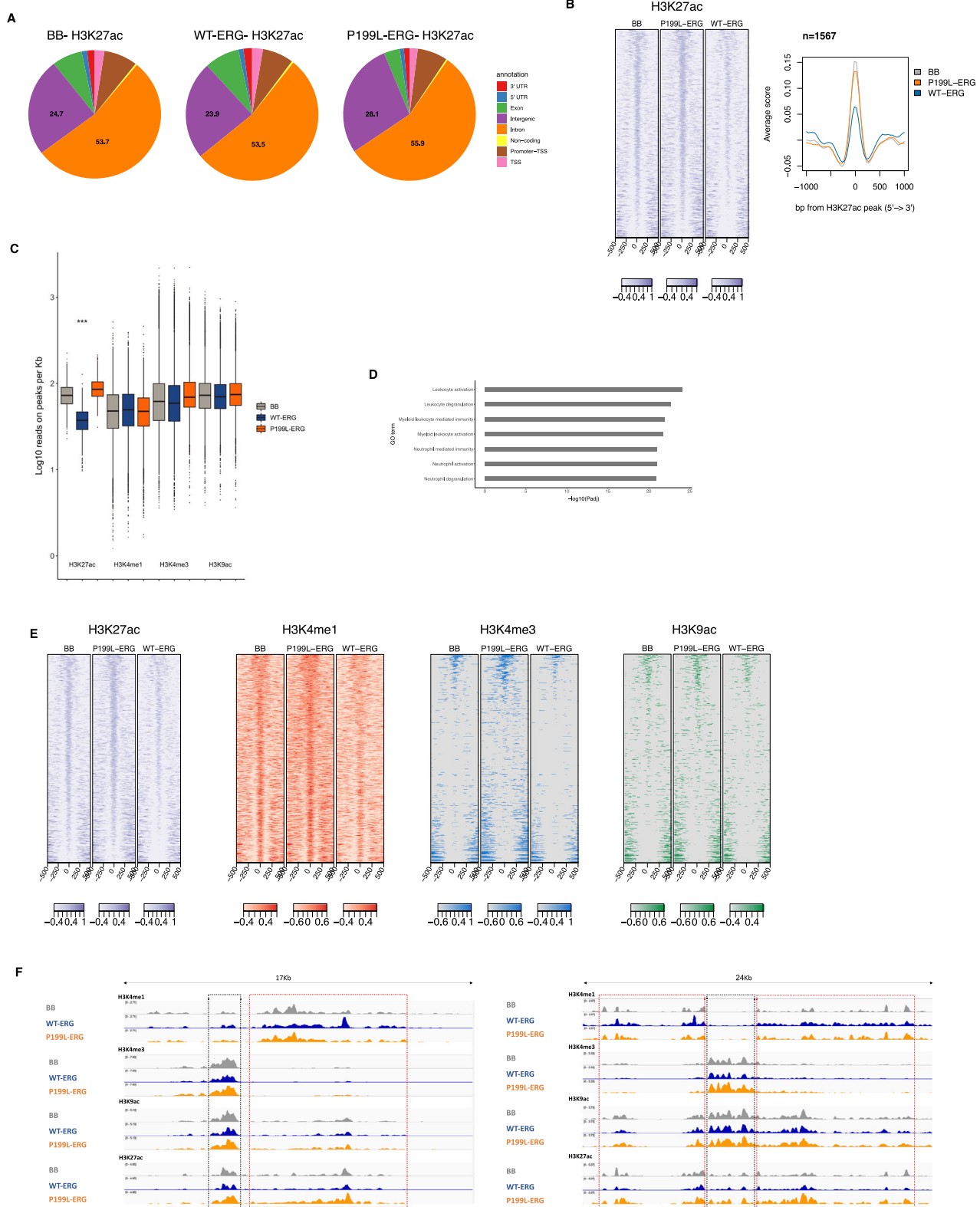

**Fig. 6 | ERG affects chromatin modifications at myeloid-related gene loci.**
**A** Global annotation statistics on the enrichment of ChIP signal of H3K27ac across groups. **B** Heatmap hierarchical clustering (left panel) and metagene plots (right panel) showing H3K27ac peaks of downregulated and hypoacetylated DEG following expression of WT-ERG force in Hoxb8 cells. **C** The intensity signal of normalized read counts of histone peaks is presented for genes identified in **B** (unpaired Wilcoxson test, *P*-value = 2.2 × 10⁻¹⁶). **D** GO analysis based on the nearest transcription start site to peaks identified in B. Adjusted *P*-values are displayed. **E** Heatmap hierarchical clustering centered on H3K27ac peaks associated with genes identified in **B**. The co-occupancy for H3K4me1, H3K4me3 and H3K9ac marks is shown. **F** Track example for two genes identified in **B** and **D**—Ly6c2 (Gr1) and Clec12a. Dashed black boxes indicate regions of H3K4me1 and H3K27ac overlap, and dashed red boxes indicate regions that relate to the transcription start site (H3K4me3, H3K9ac, and H3K27ac overlap). Source data are provided as a Source Data file.

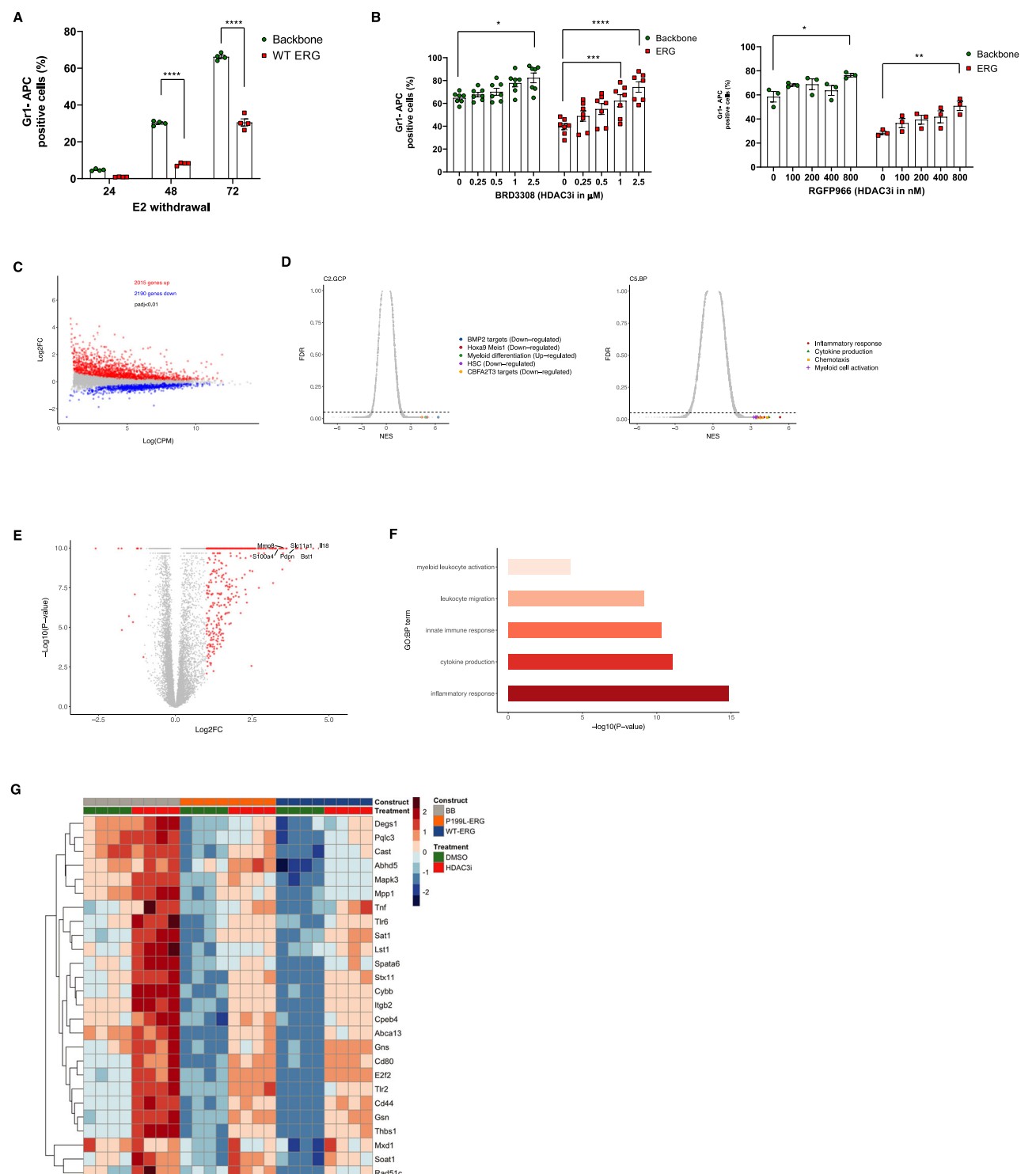

RUNX1T1 translocation and ELF-153 complex karyotype AML, which are both addicted to ERG expression[25,50] (Fig. 8B) and dependent on HDAC3 activity (Fig. 9A). NSG mice were treated intraperitoneally with 15 mg/kg RGFP966 for 21 days 24 h following transplantation with $2.5 \times 10^5$ SKNO1 and $10^6$ ELF-153 cells. RGFP966 exhibited a significant anti-leukemic effect as measured by the burden of disease in the bone marrow (Fig. 9C, D). This confirmed the importance of small molecule HDAC3 inhibition on the growth of SKNO1 and ELF-153 cells in vivo.

As pharmacological inhibitors may induce off-target effects, we validated the importance of HDAC3 using genetic tools. It was previously shown that the KO of *Hdac3* in *Vav-Cre* transgenic mice led to

dramatic defects in the growth of early hematopoietic precursors and failure to engraft in lethally irradiated mice[51]. We therefore aimed to decrease the expression of HDAC3 in SKNO1 cells (rather than a complete KO) due to concerns about cell death and poor engraftment following KO of HDAC3. The CRISPR-dCAS9 system was used in SKNO1 cells to decrease HDAC3 expression, followed by transplantation into NSG mice (Fig. S16A). Strikingly, leukemia development was abrogated in mice transplanted with SKNO1 cells with reduced expression of HDAC3 (follow-up of 45 days) (Fig. 9E). This effect was translated into a significant survival advantage after a longer follow-up period of 165 days (Fig. 9F). In contrast, HDAC3 inhibition in THP1 cells had no negative effect on leukemia development in vivo (Fig. 9G and

**Fig. 7 | HDAC3 inhibition alleviates the myeloid differentiation block induced by ERG forced expression in hematopoietic progenitor cells. A** Analysis of myeloid differentiation in human ERG-expressing ER-Hoxb8 cells. GR-1 percentage was measured using flow cytometry after withdrawal of beta-estradiol (E2). $n = 10,000$ cells were analyzed using flow cytometry for each biological independent sample for $n = 4$ independent experiments, data is represented as mean ± SEM, Two-way ANOVA, and Sidak's multiple comparison test. The $P$-value comparing BB vs. WT-ERG at 48 and 72 h after E2 withdraws are $P < 0.0001$. **B** Analysis of myeloid differentiation of ER-Hoxb8 cells expressing human ERG following inhibition of HDAC3 with BRD3308 (left panel) and RGFP966 (right panel). $n = 10,000$ cells were analyzed for using flow cytometry for each biological independent sample for 7 independent experiments for BRD3008, $n = 3$ independent experiments for RGFP966. Data is represented as mean ± SEM for both BRD3008 and RGFP966, Two-way ANOVA, and Tukey's multiple comparison test. The $P$-value comparing Backbone DMSO vs. Backbone BRD3308 (2.5 μM) is 0.0165. The $P$-value comparing ERG DMSO vs. ERG BRD3308 (1 μM) is 0.008. The $P$-value comparing ERG DMSO vs.

ERG BRD3308 (2.5 μM) is <0.0001. The $P$-value comparing Backbone DMSO vs. Backbone RGFP966 (800 nM) is 0.0169. The $P$-value comparing ERG DMSO vs. ERG RGFP966 (800 nM) is 0.0023. **C** RNA sequencing in human ERG-expressing ER-Hoxb8 cells treated with either DMSO or 2.5 μM BRD3308. An MA plot illustrates the effect of BRD3308 treatment on gene expression. Differentially expressed genes ($P_{adj} < 0.05$) are indicated. **D** GSEA of genetic and chemical perturbation (C2.CGP) and gene ontology biological process gene sets (C5.BP) following HDAC3 inhibition in human ERG-expressing ER-Hoxb8 cells. Most significant upregulated gene sets are indicated. **E** Most significant differentially expressed genes following HDAC3 inhibition in human ERG-expressing ER-Hoxb8 cells. Genes in red indicate log2 fold-change >1 and < − 1 and FDR < 0.01. **F** Gene Ontology on differentially expressed genes presented in **E**. **G** Gene expression matrix of myeloid differentiation-related genes in ER-Hoxb8 cells expressing ERG variants. DEG are presented ($P_{adj} < 0.05$). CPM−counts per million. NES−Normalized enrichment score. FDR−false-discovery rate. DEG−differentially expressed genes. ns−non-significant, * denotes $P \leq 0.05$, ** denotes $P \leq 0.01$, *** denotes $P \leq 0.001$, **** denotes $P \leq 0.0001$.

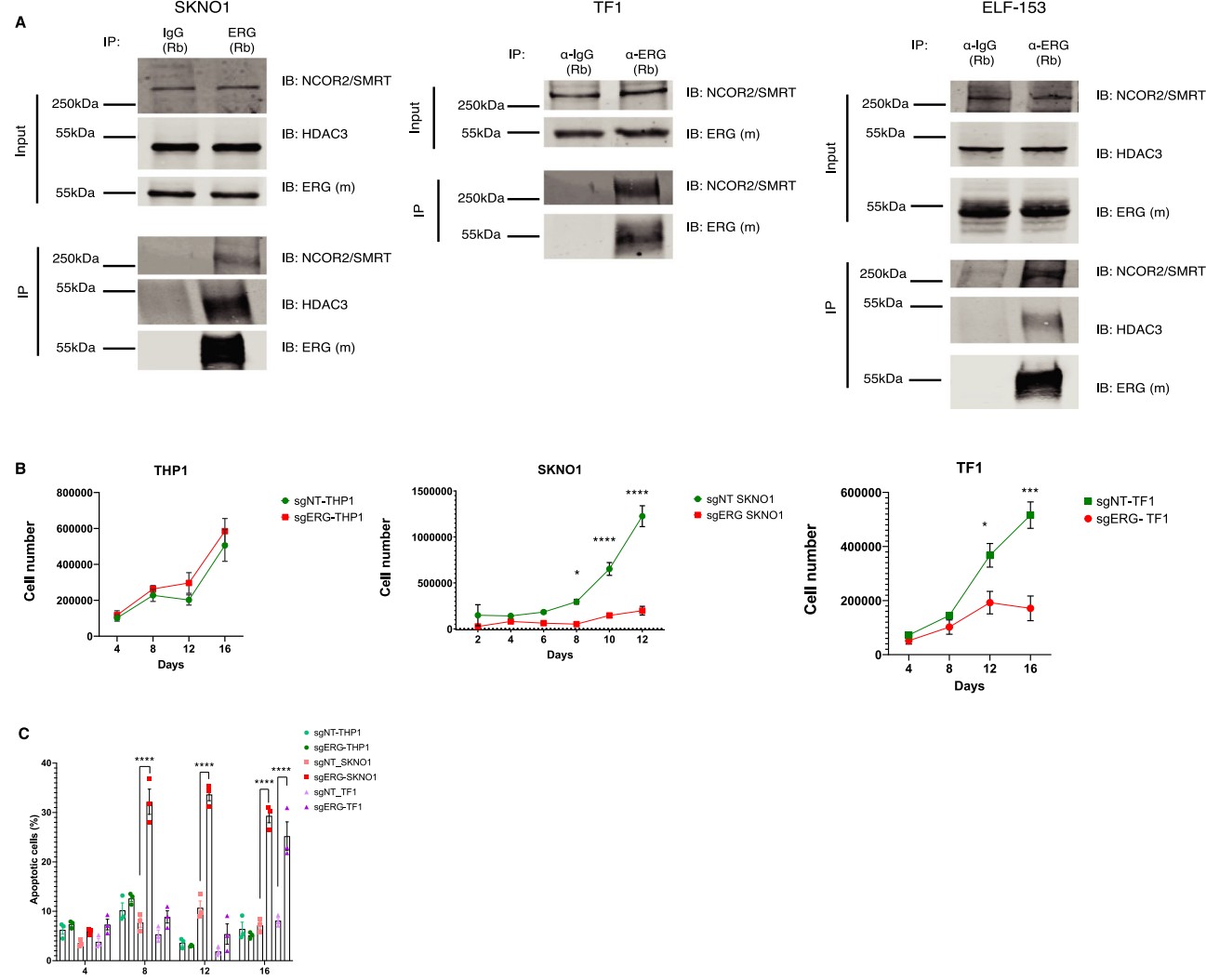

**Fig. 8 | Analysis of ERG dependency in leukemia cells with differential ERG expression. A** Co-IP assays in ERG-dependent AML cell lines. IP for endogenous ERG and blot for NCoR2/HDAC3. A representative figure of three independent experiments is shown. **B** Cell growth curves following ERG knockout using CRISPR-Cas9 system. Cells were counted by flow cytometry with a fixed volume of 60 μL in each condition out of 1 mL sample. $n = 3$ independent experiments in each experiment for each biological condition $n = 1$ biological independent sample. The $P$-values comparing sgNT-TF1 vs. sgERG-TF1 at days 12 and 16 are $P = 0.008$ and

$P < 0.0001$, respectively. The $P$-values comparing sgNT-SKNO1 vs. sgERG-SKNO1 at days 8, 10, and 12 are $P = 0.0225$, $P < 0.0001$, and $P < 0.0001$ respectively. **C** Assessment of the degree of apoptosis as a function of cell type and days in culture. $n = 10,000$ cells were counted using flow cytometry over 3 independent experiments in each experiment $n = 1$ independent biological sample. Data are represented as mean ± SEM, two-way ANOVA, Tukey's multiple comparison test. The $P$-values comparing sgNT-SKNO1 vs. sgERG-SKNO1 at days 8, 10, and 12 are $P < 0.0001$. The $P$-value comparing sgNT-TF1 vs. sgERG-TF1 at 16 is $P < 0.0001$.

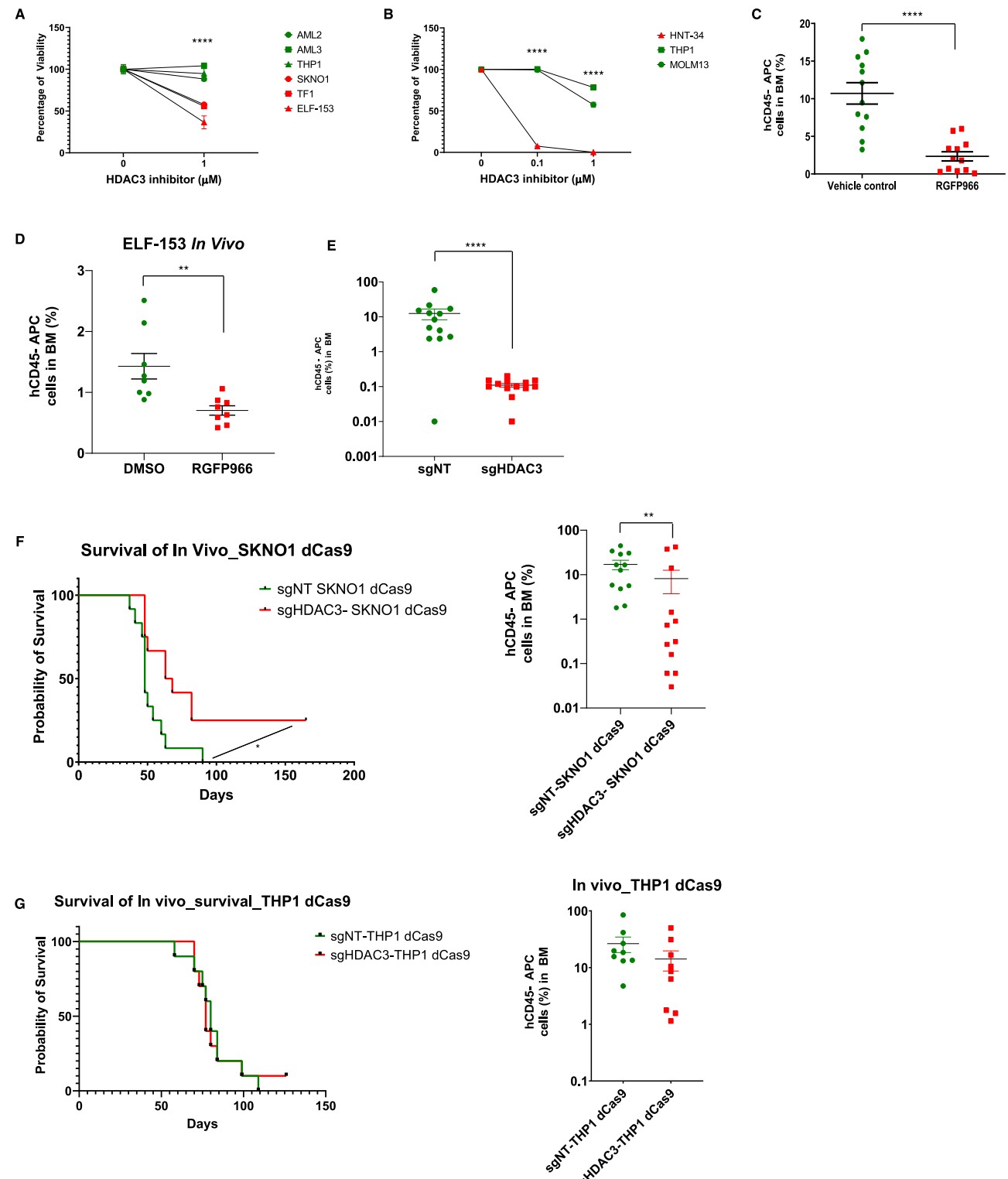

Fig. S16B). Taken together, these results suggest that HDAC3 targeting in ERG-dependent AML has an anti-leukemic effect.

## Discussion

ERG is a member of a core set of stem cell-specific genes expressed in both normal and leukemic stem cells and is linked to poor prognosis in AML. The molecular mechanisms by which ERG contributes to the development of AML are largely unknown. Here, we identified a single proline at the end of the PNT domain that is required for ERG leukemogenic function. Functional, genetic, and biochemical studies revealed that P199 contributes to the ERG-NCoR-HDAC3 interaction, thereby highlighting HDAC3 as a potential therapeutic target for ERG-mediated leukemias and, possibly, other malignancies such as prostate cancer.

The physical characteristics of the ERG-NCOR-HDAC3 interaction are yet to be fully understood. Whether the interaction is direct or indirect, and the possible involvement of additional binding proteins, remains unknown. Nevertheless, our study aimed to elucidate the functional aspects of this interaction. Specifically, we show that his interaction is important for ERG-mediated leukemogenesis, in particular for the essential step of differentiation arrest.

**Fig. 9 | Pharmacologic and genetic HDAC3 inhibition results in reduced cell growth in high ERG-expressing human AML cells in vitro and in vivo. A** HDAC3 inhibition using RGFP966 in low and high ERG-expressing AML cell lines. Mean number of cells relative to RGFP966 concentration 0 after 48 h in culture. $n = 3$ independent experiments in each experiment for each biological condition $n = 1$ biological independent sample, Data are represented as mean ± SEM, two-way ANOVA, Tukey's multiple comparison test. The $P$-value comparing ERG-dependent vs. ERG-independent with HDAC3 inhibitor is $P < 0.0001$. **B** HDAC3 inhibition using RGFP966 in HNT-34 cells. Cells were counted by flow cytometry with a fixed volume of 60 μL in each condition out of 1 mL sample. $n = 3$ independent experiments in each experiment for each biological condition $n = 1$ biological independent sample, two-way ANOVA, Tukey's multiple comparison test. The $P$-value comparing HNT34 against THP1 and MOLM-13 at 0.1 μM and 1 μM of RGP966 is $P < 0.0001$. **C** The effect of HDAC3 inhibition on SKNO1 cells in vivo. In one biological experiment, $n = 12$ animals in the treatment group (vehicle control or RGFP966), mean ± SEM, t-test, unpaired, Mann–Whitney. The bone marrow percentage of human CD45 cells was used to assess disease burden. The $P$-value comparing Vehicle control vs. RGFP966 group is $P < 0.0001$. **D** The effect of HDAC3 inhibition on ELF-153 cells in vivo. In one biological experiment, $n = 7$ animals in the vehicle group while $n = 8$ animals in the

RGFP966 group, mean ± SEM, t-test, unpaired, Mann–Whitney. The bone marrow percentage of human CD45 cells was used to assess disease burden. The $P$-value comparing Vehicle control vs. RGFP966 group is $P < 0.0011$. **E** Inhibition of HDAC3 using CRISPR-dCas9 system in SKNO1 cells. The bone marrow percentage of human CD45 cells was used to assess disease burden. In one biological experiment, $n = 12$ animals in the treatment group (sgNT or sgHDAC3), mean ± SEM, t-test, unpaired, Mann–Whitney. The $P$-value comparing sgNT vs. the sgHDAC3 group is $P < 0.0001$. **F** Survival curve of NSG mice transplanted with SKNO1 cells after CRISPR-dCas9 targeting of HDAC3 (left panel, log-rank test, $P$-value = 0.0105). The percentage of bone marrow hCD45 cells as a measure of disease burden (right panel, mean ± SEM, t-test, unpaired, Mann–Whitney). In one biological experiment, $n = 12$ animals in the treatment group (sgNT or sgHDAC3). The $P$-value comparing sgNT-SKNO1 dCas9 vs. sgHDAC3-SKNO1 dCas9 group is $P = 0.0098$. **G** Survival curve of NSG mice transplanted with THP1 cells after CRISPR-dCas9 targeting of HDAC3 (left panel, $n = 10$ animals in each group). The percentage of bone marrow hCD45 cells as a measure of disease burden (right panel, flow cytometry analysis for $n = 9$ animals in each group, Wilcoxon test). In one biological experiment, $n = 10$ animals in the treatment group (sgNT or sgHDAC3). sgNT indicates non-targeting guide, * denotes $P \leq 0.05$, ** denotes $P \leq 0.01$, *** denotes $P \leq 0.001$, **** denotes $P \leq 0.0001$.

P199L is not a complete loss of function mutation. Similar to WT-ERG, it induces stem cell (cKit, Sca1) and megakaryocyte surface marker (CD41, CD61) expression and promotes an erythroid to megakaryocytic phenotype switch in murine HSPC and K562, respectively. In murine fetal liver-derived HSPC, forced expression of P199L-ERG restricted myeloid differentiation in comparison with controls, although not at the same level as WT-ERG. Furthermore, both RNA sequencing and the ERG stem cell enhancer reporter assays in 293T and SKNO1 cells have indicated that P199L-ERG does not have a dominant negative effect on endogenous Erg. Together, our results indicate that P199L-ERG is functional, but not to the same extent as WT-ERG. As shown in the proximity map of ERG variants and in RNA sequencing of HSPC, the effect of P199L on ERG is subtle. It is nevertheless sufficient to significantly inhibit leukemogenesis.

Interestingly, the phenotype of P199L-ERG is not equivalent to the phenotype of ERG with a deleted PNT domain. This indicates that the PNT domain has additional functions that are independent of those associated with the P199 site.

The failure to differentiate into mature myeloid cells is one of the hallmarks of AML.

We show that P199L negatively affects ERG-induced repression of myeloid differentiation genes and self-renewal of HSPC.

Using a transgenic mouse model of AML, we have previously identified an ERG repressive gene expression signature[13] (Fig. 1F in the original paper). The ERG repressive gene expression signatures in HSPC strongly correlated with the signature of CBFA2T3 (ETO2, MTG16) and were impaired by P199L. CBFA2T3 is known to act in conjunction with the NCoR/SMRT/HDAC3 complex to regulate the expression of target genes in HSPC and MEP, thus controlling stemness and terminal differentiation[35,36,52–54]. CBFA2T3 and ERG have also been implicated in AMKL, where ERG expression was shown to be induced by the *CBFA2T3−GLIS2* translocation and was critical for leukemic cell survival[24]. Our results show that P199L reduces the proximity-based BioID signal to the NCoR-HDAC3 complex. Taken together, these findings support the hypothesis that, similar to CBFA2T3, ERG works in concert with NCoR-HDAC3 to restrict myeloid differentiation in HSPCs. In further support of this hypothesis, HDAC3 inhibition in human ERG-expressing murine HSPCs reversed the transcriptional and phenotypic effects of ERG by de-repressing genes normally expressed in mature myeloid cells.

ERG's role as a transcriptional repressor of cell differentiation was previously identified in prostate cancer, where aberrant expression of ERG inhibits the normal androgen receptor (AR) signaling pathway in coordination with transcriptional repression complexes[55]. ChIP-sequencing demonstrated co-binding of AR, ERG, HDAC1, HDAC2, HDAC3, and EZH2 on both promoters (HDAC1-dependent) and distal

enhancer (HDAC3-dependent) to reprogram normal AR signaling and mediate transcriptional repression of cytoskeletal genes, which associate with epithelial differentiation. The de-repression of AR target genes upon treatment with a pan-HDAC inhibitor was much more significant in ERG-positive prostate cancer cells than ERG-negative prostate cancer cells[55,56]. In the present study, we specifically targeted HDAC3 in prostate cancer cells, highlighting a potential role for the ERG-NCoR-HDAC3 pathway in nonhematologic malignancies characterized by altered ERG expression. Future research should further explore this finding.

Chromatin dynamics control gene expression and lineage specification in hematopoiesis. The landscape of enhancers and super-enhancers defines cell identity better than transcriptomic profile and promoter usage[57]. Cell fate-specific enhancer establishment is initiated at the early stages of lineage commitment in progenitor cells and is marked by H3K4me1 (enhancer priming)[58]. The gain of H3K27ac is a late event during myeloid differentiation and is established together with active gene transcription once the cells are terminally differentiated. Together with MEIS1 and HOXA9, ERG was established as a regulator of hematopoietic stem cell enhancers[30,59,60]. Moreover, it was recently discovered that evolutionarily conserved heptad enhancers in AML rely on ETS motifs for their activity[61]. Our findings suggest that ERG additionally restricts the activity of myeloid-specific enhancers in HSPC, as ERG forced expression led to a decrease in the H3K27ac signature at myeloid differentiation-related gene loci associated mostly with H3K4me1. A gain in H3K27ac was detected for cells that were transduced with P199L-ERG at similar sites, thus supporting the hypothesis that the interaction with the NCoR-HDAC3 complex is essential for the process.

SKNO1 human AML cells carry the t(8;21) translocation fusing *RUNX1* with *RUNX1T1* that accounts for 7% of adult AMLs[62]. While RUNX1 is a transcription activator, RUNX1T1 was shown to act as a co-repressor by recruiting the NCoR-HDAC complex to regulatory elements of target genes[63,64]. By replacing the transactivating domain of RUNX1 with almost the entirety of RUNX1T1, the oncogene generated by t(8;21) is thought to convert the RUNX1 transcriptional activator to a strong repressor[25,65–67]. Analysis of the transcriptome and epigenome of t(8;21) patient cells revealed binding of ERG, FLI1, TAL1, and RUNX1 at all RUNX1-RUNX1T1 occupied regulatory regions. Knockdown of ERG resulted in cell death and was accompanied, surprisingly, by an increased expression of RUNX1-RUNX1T1, oncogenic overdose, and cell lethality. ChIP-qPCR after the knockdown of ERG revealed an increase in P300 and a decrease in HDAC1 occupancy at the promoter region of RUNX1-RUNX1T1[25]. Furthermore, a recent study has demonstrated that RUNX1 is a component of the NCoR-HDAC3 complex in t(8;21) AML and collaboratively represses RUNX1-RUNX1T1-

dependent transcription, thereby linking HDAC3 directly to leukemogenesis associated with t(8;21)[68]. Taken together, the enhanced apoptosis observed in SKNO1 cells following HDAC3 inhibition could be explained by a co-regulatory function of ERG and the NCoR-HDAC3 complex in t(8;21) leukemic cells. As we demonstrated, the ERG-NCoR-HDAC3 interaction may also be essential for the maintenance and progression of leukemogenesis in EVI-driven, ERG-dependent AML. Further investigation is needed to confirm this observation.

In concordance with our findings, a recent protein interactome study of cellular and chromatin-associated HOXA9 in AML supports the hypothesis that a block in myeloid differentiation is an active and crucial process for the maintenance of leukemia. The study demonstrated that HOXA9 interacts with the matrix-binding protein SAFB and recruits the NuRD remodeling complex. This complex represses gene expression, contributing to the maintenance of AML[69]. Thus, the discoveries by us and by Dr. Huntly's group suggest that interaction between leukemogenic transcription factors and repressor complexes generates the differentiation arrest typical for AML.

The development of HDAC inhibitors for the therapy of hematopoietic malignancies originated from the observation that several compounds that were known to induce the differentiation of leukemic cell lines were HDAC inhibitors[70,71]. However, the results of several clinical studies were disappointing[72,73]. A pan-HDAC inhibitor was used in these clinical studies, and thus the role of specific types of HDACs in AML is not known. HDAC3-specific inhibition was recently tested in a mouse model of AML and was shown to be a useful target in combination with cytarabine. The synergistic effect of HDAC3 inhibition and cytarabine was mechanistically attributed to the involvement of the former in the DNA damage response[45].

Our findings indicate that the aberrant overexpression of ERG maintains HSPCs in an undifferentiated state and promotes AML development. We suggest that the interaction between ERG and the NCoR-HDAC3 complex has an important role in the leukemogenic process and that HDAC3 inhibition could be beneficial in AML characterized by high ERG expression. ERG is a transcription factor and is generally not considered a druggable target. As such, the mechanism proposed here identifies HDAC3 inhibition as a potential therapeutic route for the treatment of ERG-driven and ERG-dependent myeloid and lymphoid leukemias. Our findings in ERG-mediated prostate cancer cells suggest that this approach may have a general role in ERG-driven malignancies.

## Methods

This research was carried out in strict adherence to all relevant ethical guidelines and regulations. Sheba Medical Center's Ethics Committee, including the Helsinki Committee, thoroughly reviewed the study protocol and approved it, ensuring compliance with the 1964 Helsinki Declaration and its subsequent amendments or comparable ethical standards. The procedures performed were in accordance with the ethical standards laid down by these committees and the institution as a whole.

In our study, strict adherence to ethical regulations regarding animal research was maintained at all times. Mice were frequently examined for any signs of distress, changes in behavior, appearance, or difficulty moving, all to ensure their well-being. If a mouse exhibited any of these signs or if the tumor burden reached the limit, the animal was humanely euthanized to prevent further suffering.

### Cell lines and primary cultures

All the cells used were maintained at 37 °C in 5% CO2. HEK-293T cells are human embryonic kidney cell lines used routinely for virus preparations. The cells were grown in DMEM (Gibco) supplemented with 10% FBS (Gibco), 1% L-Glutamine, and 1% penicillin/streptomycin. Estrogen-regulated ER-Hoxb8 granulocyte-monocyte murine progenitors (GMP) were kindly provided by Prof. David Sykes at the

Massachusetts General Hospital. It is a murine GMP cell line that conditionally expresses the ER-Hoxb8 oncoprotein under the regulation of the estrogen receptor. As estrogen is withdrawn from the culture media, the cells differentiate into mature neutrophils in a process that typically lasts 5 days[43]. Cells are grown in RPMI (Gibco) supplemented with 10% FBS (Gibco), 1% L-Glutamine, and 1% penicillin/streptomycin. In addition, 2% SCF and beta-estradiol (Sigma, E-2758) in a concentration of 0.5 micromolar are added to the culture media. AML2, AML3, and THP1 cells were grown in 10%RPMI. SKNO1 cells were grown in 10%RPMI with 10 ng/mL human GM-CSF. ELF-153 cells were grown in 10%RPMI with 5 ng/mL human GM-CSF. TF1 cells were grown in 20% RPMI with 5 ng/mL human GM-CSF. HNT34 cells were grown in 20% RPMI with 100 µM HEPES and 100 µM Sodium Pyruvate. LNCaP cells were grown in 10% RPMI and VCaP cells were grown in 10% DMEM with 100 µM Sodium Pyruvate and 100 µM Glutamax supplement.

Primary murine fetal liver-derived HSPC were cultured at 37 °C in 5% CO2 in serum-free medium supplemented with (Stem Cell Technologies, StemSpan SFEM, #09605), % L-Glutamine, 1% penicillin/streptomycin, mSCF (10 ng/mL, PeproTech), mTPO (10 ng/mL, PeproTech) and mFLT3 (10 ng/mL, PeproTech).

### Cell line sources

HEK-293T cells—purchased from ATCC

Estrogen-regulated ER-Hoxb8 granulocyte-monocyte murine progenitors (GMP) were kindly provided by Prof. David Sykes at the Massachusetts General Hospital.

SKNOI, AML2, AML3, THP1, TFl, ELF-153—were kindly provided by Dr. Michael Milyavsky, Tel-Aviv University.

These lines were purchased by Dr. Milyavsky from DSMZ.

THP1 was given from Dr. John Dick lab to Dr. Milyavsky.

HNT-34 and MOLM-13 were purchased and maintained by the Zuber lab.

The Bourquin Laboratory generated and maintained TCF3-HLF PDXs.

### Animals

Statistical design for experiments involving mice:

Sample size—We calculated the sample size for experiments involving mice using power analysis. We used the freely downloadable software G Power (Faul, Erdfelder, Lang and Buchner, 2007).

Data exclusion—In the immunophenotyping analysis, only GFP-positive ERG-transduced cells were analyzed.

Replication—The results in this manuscript are a summary of repeated mice experiments. In the mice experiment presented in Fig. 3A, six batches of fetal liver-derived hematopoietic stem and progenitor cells were used over a two-year period. In the Skno1 in vivo experiments, two batches were used for both experiments (HDAC3 enzymatic inhibition and CRISPR-dCAS9 targeting) over a 2-month period. There were no unsuccessful randomizations. The replication attempts were all successful for the fetal liver-derived and SKNO1 experiments.

Randomization—Each batch of fetal liver-derived hematopoietic stem and progenitor cells was split equally for several experimental groups. The control group was included in each batch. Mice were marked and transplanted randomly by the technician who was blind to experimental parameters. HDAC3 inhibition in mice transplanted with ERG-dependent and dependent cell lines was randomized in a similar manner to the fetal liver experiment.

Blinding—All transplantations were done by a technician who was blind to group allocation.

Flow cytometry data collection and analysis were performed by the first co-authors and were not blinded.

Pregnant C57Bl/6 female mice for transduction-transplantation assays were purchased from Envigo laboratory. For transplantation assays, six weeks old female C57Bl/6 mice were included in the study.

Mice were housed in a barrier facility. Animal studies were approved by Tel-Aviv University and Chaim Sheba Medical Center at Tel Hashomer Institutional Animal Care and Use Committees. The Helsinki ethics committee at the Chaim Sheba Medical Center authorized all in vivo experiments (authorization number: 14-051M) for up to 800 mice during the years 2014-2018.

## Vectors and mutagenesis

Human cDNA of *ERG* was cloned to MA-1 bidirectional second-generation lentivirus expression vector[74]. The expression of *ERG* is regulated under the hPGK promoter, and the expression of GFP is regulated under the minimal CMV promoter. cDNA of human *ERG* was cloned to MA-1using Xba1 (5′) and Sma1(3′) restriction sites. The insert contained a Kozak sequence and an HA tag (5′). Site-directed mutagenesis was carried out to introduce the P199L mutation. Site-directed mutagenesis was carried out using the QuikChange II Site-Directed Mutagenesis Kit (catalog #200523) according to the manufacturer's instructions.

## Enrichment of mouse hematopoietic stem and progenitors

Pregnant mice (purchased from Envigo laboratory) were sacrificed on E13.5 and fetal liver cells were isolated. HSPC was enriched using the lineage-positive depletion magnetic kit (Miltenyi Biotec MACS Lineage cell depletion kit,130-090-858). Lineage depleted cells were cultured at 37 °C in 5% $CO_2$ in serum-free medium supplemented with (Stem Cell Technologies, StemSpan SFEM, #09605), % L-Glutamine, 1% penicillin/ streptomycin, mSCF (10 ng/mL, PeproTech), mTPO (10 ng/mL, PeproTech) and mFLT3 (10 ng/mL, PeproTech).

## Lentiviral experiments

Lentiviral supernatants were generated by co-transfection of 293T cells with MA-1 as the expression vector and viral packaging plasmids psPAX2 and pMD2.G (Addgene ID 12260 and 12259 respectively). Cells were transfected using the calcium phosphate Profection Mammalian Transfection kit (Promega) according to the manufacturer's instructions. The supernatant was collected twice: at 24 and 48 h following transfection and filtered through a 0.45 μm strainer. The supernatant was concentrated using Vivaspin (Sartorius) for the transduction of ER-Hoxb8 cells. For the transduction of fetal liver-derived HSPC, the supernatant was concentrated using ultracentrifuge (3:30 h, 20,000 rpm, 10 °C).

To transduce murine fetal liver-derived HSPC, we cultured the primary cells in 96-well plates with serum-free medium conditions as described. Each well was cultured with 50,000 cells. The viral titer was aimed to be approximately 50 MOI. Five μg/mL Polybrene was added to each well. We used spinfection for transduction at a rate of $1000 \times g$ for 90 min at 22 °C. After centrifugation, the cells were incubated at 37 °C in 5% $CO_2$. Forty-eight hours post-transduction, the cells were collected, washed, and used for further applications. The transduction efficiency was measured using flow cytometry for GFP.

## Methylcellulose re-plating assays

Murine fetal liver-derived HSPC were transduced with MA-1 Lentiviruses expressing either WT-ERG, P199L-ERG, or an empty vector as a control. Ten thousand cells per construct were plated in duplicates of 1 mL methylcellulose supplemented with mIL-3, mIL-6, mSCF (Stem Cell Technologies mouse MethoCult M3534). Cells were incubated at 37 °C in 5% $CO_2$ for seven to 10 days. Following incubation, colonies consisting of more than 50 cells were counted, and 10,000 cells were re-plated in duplicates in new methylcellulose cultures. GFP verified the presence of transduced cells.

## Transduction-transplantation assays

Murine fetal liver-derived HSPC were enriched and transduced with expression vectors carrying ERG variants as described above. The transduction efficiency of each construct was evaluated by GFP percentage using a Galios Flow Cytometer (Beckman-Coulter).

Six-week-old C57Bl/6 mice female recipient (purchased from Envigo) were administered a sublethal dose of 650 rad X-ray irradiation 24 h prior to transplantation. Recipient mice received prophylactic antibiotic treatment 48 h before transplantation and until 48 h post-transplantation. 105 transduced cells together with $2 \times 10^5$ BM support cells (freshly harvested from the bone marrow of C57Bl/6 mice and lysed for red blood cells) were resuspended in PBS and injected via the tail vein of irradiated recipients.

## ER-Hoxb8 differentiation assay

Stable cell lines of ER-Hoxb8 cells overexpressing *ERG* variants were created.

For viral transduction, 200,000 cells were cultured in 6-well plates in RPMI supplemented with 10% FBS (Gibco), 1% L-Glutamine, and 1% penicillin/streptomycin, 2% SCF and 0.5UM beta-estradiol as described above. Cells were transduced with a lentiviral containing expressing vectors of WT-*ERG*, P199L-*ERG*, and an empty vector. Lentiviral supernatant was added to each plate together with 5 μg/mL Polybrene. Cells were centrifuged for $1000 \times g$ for 90 min at 22 °C and were incubated at 37 °C in 5% $CO_2$. Forty-eight hours following spinfection, the transduction rate was measured using GFP and was generally 20–30%. GFP-positive cells were then sorted and incubated at 37 °C in 5% $CO_2$ for 1 week. As cells recovered from the sorting procedure and proliferated, 50,000 ER-Hoxb8 cells were carefully washed to remove the beta-estradiol remaining and re-cultured in 24-well plates in the ER-Hoxb8 medium as described above without beta-estradiol. We next assessed the differentiation rate of the cells towards mature granulocytes as a function cell construct. Flow cytometry was carried out using an antibody for Gr1 (APC, Biolegend). Mean fluorescence intensity was measured for comparison.

## Flow cytometry

Flow cytometry was used in determining transfection/ transduction efficiency, cell count, and for immunophenotyping. For all the above, cells were washed in staining media (2%FBS in PBS) and resuspended in 100 μL staining media containing fluorochrome-conjugated antibodies for 30 min. Following staining, cells were washed with staining media, resuspended to a final volume of 100 μL staining media, and analyzed on a Gallios flow cytometer (Beckman-Coulter). All cell mixtures were stained with an appropriate color of Fixable Viability Dye (Life Technologies) to exclude dead cells from analysis. Quadrant gates were defined according to single stains and full minus one staining (FMO) of each fluorophore. Data were analyzed with FlowJo software (BD).

The following antibodies were used in this study for flow cytometry:

Differentiation analysis of human leukemia cell lines upon HDAC3 inhibition, Dilution 1:100:

 CD13 PE BD 347406
 CD34 PerCPCy5.5 BD 347222
 CD117 PECy7 BD 333950
 CD33 APC BD 345800
 HAL-DR PB450 Biolegend 307633
 CD45 KO500 BC B36294
 CD36 FITC BC IMO766U
 CD64 PE BC IM3601U
 CD14 APC H7 BD 641394

Propagation of human leukemia cell lines in immune-deficient mice:

 hCD45 #17-0459-42, APC, Bioscience (thermofisher).

Fetal liver-derived HSCP−immune phenotype:

Alexa Fluor® 700 anti-mouse Ly-6A/E (Sca-1) Antibody, clone D7, cat # 108141, Biolegend

PE/Cyanine7 anti-mouse CD117 (c-Kit) Antibody, clone 2B8, cat# 105813,Biolegend

APC anti-mouse CD150 (SLAM) Antibody, Clone W19132B cat # 162603, Biolegend

Pacific Blue™ anti-mouse CD150 (SLAM) Antibody, clone TC15-12F12.2, cat# 115923, Biolegend

PE anti-mouse CD41 Antibody clone MWReg30 cat# 133905, Biolegend

APC anti-mouse CD61 clone 2C9.G2 (HMβ3-1) Antibody cat# 104315, Biolegend

PE anti-mouse/human CD11b Antibody, clone M1/70, cat # 101207, Biolegend

APC anti-mouse Ly-6G/Ly-6C (Gr-1) Antibody, clone RB6-8C5, cat # 108411, Biolegend

PE anti-mouse CD34 Antibody Clone SA376A4, cat # 152203. Biolegend

APC Annexin V, cat # 640919, Biolegend

## Western blot analysis

In all, $2 \times 10^6$ cells were lysed using either RIPA buffer or CellLytic M (Sigma) with a HALT protease inhibitor cocktail (Thermofisher). Protein lysates were separated on SDS gel and transferred to a nitrocellulose membrane. Membranes were blocked for 1 h with PBS-T containing 5% skim milk and were incubated at 4 °C overnight with a primary antibody. Membranes were washed with PBS-Tween (0.05%) and incubated with a secondary antibody (1:10,000, Jackson ImmunoResearch Laboratories) for 2 hr at room temperature. The membrane was rewashed with PBS-T. Proteins were detected using enhanced chemiluminescence (ClarityTM Western ECL Blotting Substrates Bio-rad), and signals were detected using a gel documentation system (Bio-rad). For ERG expression and cellular sub-localization: 293T cells were transiently transfected with *ERG* expression vectors using the calcium phosphate Profection Mammalian Transfection kit (Promega). 48 hours following transfection, cells were collected, and cytoplasmatic or nuclear proteins were extracted using NucBuster Protein Extraction Kit (Novagen) according to the manufacturer's instructions.

For ERG immunoblot we used anti-human ERG antibody by Santa Cruz (C-17, sc-354, 1:1000). For HA immunoblot we used anti-HA monoclonal antibody (Sigma, H3663). For Vinculin immunoblot we used anti-Vinculin antibody manufactured by Abcam.

## RNA sequencing

Murine fetal liver-derived HSPC was transduced with ERG variants as described above. All samples were sorted for GFP-positive cells ($5 \times 10^5$ cells/sample) by BD FACSAria, centrifuged (10 min, 1200 rpm), fluidized in TRIzol reagent, and purified using the TRIzolTM Plus RNA Purification Kit (Invitrogen). Sequencing and analysis were performed in the INCPM of Weizmann Institute of Science (The Crown Genomics Institute of the Nancy and Stephen Grand Israel National Center for Personalized Medicine, Weizmann Institute of Science, Israel). RNA was purified, and analyzed by bioanalyzer (Agilent 2100), and cDNA libraries were prepared using INCPM mRNA-seq. Genome-wide expression profiles were obtained by sequencing of the samples on Illumina HiSeq 2500 machine, SR60_V4. The output was approximately 24 million reads per sample. Resulting reads shorter than 30 bp were discarded. Reads were mapped to the M. musculus reference genome GRCm38 using STAR. Differentially expressed genes were identified using DESeq2 in R software[75]. Raw *P*-values were adjusted for multiple tests using the procedure of Benjamini and Hochberg.

## Gene-set enrichment analysis (GSEA)

Gene-set enrichment analysis was performed using a pre-ranked list[76]. All transcripts per sample with more than 30 reads were included in the analysis. The score for each gene was calculated as $\log_{10}$(*P*-value). Genes that were downregulated had a negative score and upregulated genes received a positive score. The Gene expression profile was compared to the MSigDB database (Broad). An adjusted *q*-value of less than 0.05 was used as a cutoff for significance. Analysis was conducted using the R package ClusterProfiler[77].

## Functional enrichment analysis

Over-representative analysis using g: Profiler was conducted for differentially expressed genes (adjusted *P*-value < 0.05) in ER-Hoxb8 cells overexpressing ERG variants. The gene expression signature was compared between cells treated with an HDAC3 inhibitor to DMSO-treated cells. A significance threshold of the adjusted *P*-value of 0.05 was used according to the g: SCS algorithm[78].

## ChIP sequencing

ChIP material was prepared from ER-Hoxb8 cells stably expressing ERG variants and an empty vector for control. $10^7$ cells were used per sample. Samples preparation and immunoprecipitation were conducted with the assistance of our collaborator Dr. Julie Thomas (Prof. John Pimanda's lab in Australia) and as previously described[13].

Briefly, cells were crosslinked with 1% paraformaldehyde for 15 min and were quenched with glycine for 5 min at room temperature. Lysis buffer with protease inhibitor was gently added to the cell pellet. Fixed chromatin was sonicated with the Bioruptor Pico using pre-optimized conditions and immunoprecipitated with the indicated antibody. Immunoprecipitation was performed using polyclonal antibodies raised against H3K27ac, H3K4me1, H3K4me3, and H3K9ac (Abcam). As a control, non-specific rabbit IgG (I5006; Sigma-Aldrich) was used.

For sequencing and analysis: ChIP samples were amplified and sequenced using the Illumina HiSeq2500 machine, SR60_V4 (BGI). The median sequencing depth was ~25 million reads per sample. Adapters were trimmed using the Cutadapt tool. Following adapter removal reads that were shorter than 30 nucleotides were discarded. Reads alignment and peak calling were performed at the INCPM of Weizmann Institute of Science (The Crown Genomics Institute of the Nancy and Stephen Grand Israel National Center for Personalized Medicine, Weizmann Institute of Science, Israel). The reads were aligned uniquely to the mouse genome (mm10) using bowtie (version 1.0.0). Bound regions were detected using MACS2 (version 2.0.10.20131216). GREAT was used for assigning genomic regions to genes. HOMER (version 4.7) was used for obtaining statistics on ChIP enrichment of genome features.

Heatmaps and metagene plots of the genes with changed H3K27ac read coverage were plotted using the R package "Genomation"[79]. g: Profiler was used for functional enrichment analysis[78].

## BioID and mutated ERG stability assays

**Cloning of ERG plasmids.** All ERG constructs were subcloned from a mammalian expression MA-vector (described above) encoding for an N-terminal HA tag wild type and P199L ERG. The MA-vector was used as a template for PCR to generate ERG inserts. ERG PCR inserts were cloned by restriction enzyme digestion followed by T4 ligation into the corresponding destination vector. For bacterial expression, ERG 1-265 WT or P199L were cloned into the custom vector PSJ5. PSJ5 encodes for a fusion protein with an N-terminal TrxA−6xHis−thrombin cleavage site−STag−enterokinase site−8xHis−TEV cleavage site. For BioID, ERG 1-479 WT or P199L were cloned into a piggyBac vector encoding a FLAG-BirA* N -terminal fusion protein.

**Protein expression.** PSJ5 expression plasmids were transformed into *E. coli* BL21-CodonPlus cells and selected on LB agar plates containing 100 μg/mL ampicillin. Starter cultures were grown overnight at 37 °C. Expression cultures were inoculated with a 1:40 volume ratio of starter

culture in 0.5x TB medium supplemented with 100 µg/mL of ampicillin or 50 µg/mL of kanamycin. Cells were grown at 37 °C to an $OD_{600}$ of 0.8–1.0 before inducing protein expression with 0.5 mM IPTG for 16–18 h at 15 °C. Pellets were resuspended in a Lysis Buffer (500 mM NaCl, 25 mM Tris pH 8, 1 mM βME with 400 µg of DNaseI (Sigma) and 1 cOmplete™ Protease Inhibitor Cocktail tablet) by vortexing. Cells were lysed by passage through an Emulsiflex for 15 min. The lysed cells were then centrifuged at 17,000 rpm for 1 h at 4 °C to remove insoluble material. The supernatant was then filtered using a 0.45 µM syringe filter before purification.

**Purification.** His-tagged proteins were purified from the supernatant by gravity-flow Ni-NTA affinity chromatography (Qiagen). Where indicated, affinity tags were cleaved from the fusion proteins by TEV protease treatment at a 1:30 molar ratio of protease to protein, while dialyzing in the lysis buffer for 16 h at 4 °C. Affinity tags were removed by Ni-NTA affinity chromatography and the proteins of interest were collected in the flow-through. Proteins were then concentrated and further purified by either S75 or S200 size-exclusion column chromatography in the indicated assay buffer. Peak $A_{280}$ fractions were concentrated, flash-frozen, and stored at −80 °C.

**Circular dichroism spectrum scan and temperature melt.** Proteins were dialyzed into CD Buffer (100 mM NaCl, and 25 mM $NaPO_4$ pH 7.5). Circular Dichroism (CD) experiments were performed on the Jasco J-810 Spectropolarimeter. Protein samples were diluted to 0.3 mg/mL and loaded into a 0.1 cm path length curvette. Spectrum scans were collected between 195–250 nm with a response of 8 s/nm and averaging 5 readings per wavelength. The temperature range for thermal melts ranged from 20 to 95 °C collecting at 226 nm. Data was processed using Excel. Buffer only was used as a background control. Secondary structure content was estimated using a CD-spectra simulator and fitted to the experimental spectra in Excel (Abriata 2011).

**Analytical size-exclusion chromatography.** Analytical size-exclusion chromatography (SEC) was performed on a Waters Breeze 2 HPLC with a Shodex KW-803 size-exclusion column. Samples were loaded in 110 µL volumes running at 0.8 mL/min in 150 mM NaCl, 25 mM Tris pH 8, and 0.5 mM TCEP. Elution profiles were measured at 230 and 280 nm.

**Stable cell line generation for BioID.** Stable cell lines for BioID were generated in HEK293 GnT −/− cells. Cells were transfected using Lipofectamine 2000 according to the manufacturer protocol with a few changes. A total of 1 µg of DNA was used for transfection. The DNA mixture consisted of FLAG-BirA*-ERG 1-479 WT/P199L or FLAG-BirA* only control: PBase: PB-RB at an 8:1:1 molar ratio respectively. Transfected cells were selected by drug resistance to puromycin at 1 µg/mL and blasticidin at 0.5 µg/mL for 3 weeks.

**Bait expression and biotin labeling.** Drug-resistant cells expressing inducible FLAG-BirA*-ERG 1-479 WT/P199L or FLAG-BirA* only control were expanded to five 150 mm plates. Cells were grown to 80% confluency and bait expression was induced with 1 µg/mL of doxycycline supplemented with 50 µM biotin for in vivo biotinylation for 24 h. The media was removed, and cells were washed with PBS and flash-frozen at −80 °C.

**BioID proteomics.** Cell pellets were thawed and resuspended in 10 mL of modified RIPA lysis buffer (50 mM Tris-HCl pH 7.5, 150 mM NaCl, 1 mM EDTA, 1 mM EGTA, 1% Triton X-100, 0.1% SDS, 1:500 protease inhibitor cocktail (Sigma-Aldrich), 1:100 Benzonase nuclease (Sigma-Aldrich)) at 4 °C (Coyaud et al.[80]). Cells were lysed by sonication for 30 s at 35 % power. Insoluble material was removed by centrifuging the lysate at 16,000 rpm for 30 min. The clarified supernatant containing the bait

protein was then incubated with 30 µL of streptavidin-Sepharose beads (GE) at 4 °C for 3 h. The beads were washed six times with 50 mM ammonium bicarbonate pH 8.3 to remove non-specific binders. Peptides were generated by TPCK trypsin (Promega) digestion at 37 °C for 16 h. The flow-through containing the tryptic peptides was lyophilized and resuspended in 0.1% formic acid for LC-MS/MS.

**Mass spectrometry.** LC-MS/MS experimental procedures were similar to those used in Coyaud et al.[80]. Briefly, samples were analyzed on a hybrid LTQ-Orbitrap velos mass spectrometer (Thermo Fisher Scientific) coupled to a Proxeon EASY-nLC pump operating at 250 nl/min with a 120 min reversed-phase buffer gradient at 40 °C. Orbitrap parent ion scans at a resolving power of 60000 were used to select up to twenty of the most intense parent ions for MS/MS using standard CID fragmentation and detection in the LTQ. Protein identification was carried out with X!Tandem (Craig and Beavis[81]; Kessner et al.[82]) against Human RefSeq Version 45 with a tolerance of 15 ppm for the parent ion and an MS/MS fragment ion tolerance of 0.4 Da, with up to two missed trypsin cleavages. Lysine ubiquitylation and methionine oxidation were allowed as variable modifications.

**Data analysis.** The raw data was processed using the CRAPome server (http://crapome.org). Virtual Crapome controls from total HEK293 BirA*-FLAG extracts were used in conjunction with background controls. For SAINTexpress scoring, only the top 6 average controls were considered. The output file was imported to Microsoft Excel for filtering. The filtering scheme involved excluding proteins that had bFDR scores of >0.05 and SAINT scores of <0.8. WT-ERG associations were then categorized based on molecular function using ShinyGO (http://bioinformatics.sdstate.edu/go/). Evidence for protein associations between the identified interactors was scored using STRING (https://string-db.org/). The combined score used a weighted combination of the following scores: o-occurrence score of the phyletic profile derived from similar absence/presence patterns of genes, co-expression derived from similar patterns of mRNA expression measured by DNA arrays and similar techniques, an experimental score derived from experimental data from CoIP, BIND, and IntAct. Database scores were derived from curated data from Biocarta, BioCyc, GO, KEGG, and Reactome. Top genes were manually cross-referenced with literature.

ERG WT and mutant data sets were combined to assess the changes in the interactome between P199L mutant and wild-type. The FCA ratio is calculated as the mutated/WT ratio and ranked as follows: proteins that score greater than 1.5 were strongly more abundant in the WT, proteins with scores between 1.5 and 1.2 were mildly more abundant, proteins with FCA ratios between 1.2 and 0.8 were considered relatively unchanged. Proteins that scored between 0.8 and 0.5 were considered mildly reduced in the mutant and proteins that scored below 0.5 were considered strongly reduced in the mutant. Significant changes between the WT and mutant associations were quantified using a two-tailed distribution with two sample unequal variance (heteroscedastic) t-test. Interaction

**Cell lysis, immunoblotting, and co-immunoprecipitation assays**
Fifty microliters of Dynabeads were washed three times with ice-cold HNTG buffer (20 mM HEPES, pH 7.5, 150 mM NaCl, 1% Triton X-100, and Glycerol 10%). After the beads had been washed, antibodies (Rabbit Isotope or Anti-human ERG rabbit (ab133264)) were diluted in lysis buffer and beads were incubated with these antibodies for 2 h at 4 °C. We then harvested and lysed 100 million cells using an IP lysis buffer (50 mM HEPES, pH 7.5, 10% glycerol, 150 mM NaCl, 1% Triton X-100, 1 mM EDTA, 1 mM EGTA, 10 mM NaF, and 30 mM β-glycerol phosphate). The cells were incubated on ice for 30 min for lysis, and the samples were short-vortexed every 5 min. After 30 min, samples were centrifuged at $17,000 \times g$ for 20 min to separate soluble proteins. A Bradford assay was used to determine the protein concentration in

the samples. Immediately following the incubation of antibodies with beads for 2 h, the beads were washed with ice-cold HNTG buffer to remove the unbound antibodies. Rabbit isotope antibody-conjugated beads or anti-human ERG antibody-conjugated beads were loaded with an equal amount of protein lysate. Using a rotating platform, beads conjugated with antibody and protein lysate were incubated overnight at 4 °C. The next day, the cell lysate was removed, and the beads were washed three times with ice-cold HNTG buffer. A 30 L solution of 1X Laemmli buffer was used to elute the bound proteins. An SDS-PAGE gel was run according to the manufacturer's instructions (BioRad), and a transfer was carried out according to the manufacturer's instructions (BioRad). The membrane blots were blocked with 5% BSA for 1 h at room temperature following the transfer. After 1 h, membrane blots were incubated overnight at 4 °C with anti-ERG (sc-271048), anti-NCOR2/SMRT (sc-32298), and anti-HDAC3 (sc-11795). Next, membrane blots were washed with 1x PBST three times, followed by incubation with a secondary fluorescent antibody (ab216772) at 1:20,000 dilution for 1 h. Odessy CLX was used to analyze the blots.

### HDAC3 inhibition in vitro
For inhibiting HDAC3 in vitro we used BRD3308 (Sigma, 1639) and RGF966 (Selleck). The inhibitors were dissolved in DMSO according to the manufacturer instructions.

**Differentiation assays of ER-Hoxb8 cells treated with BRD3308.** In a 24-well plate, 50,000 ER-Hoxb8 cells stably expressing ERG variants were plated after beta-estradiol was washed away from the cell culture medium (RPMI supplemented with 10% FBS (Gibco), 1% L-Glutamine, and 1% penicillin/streptomycin, 2% SCF). Cells were treated with either DMSO or BRD3308 in escalating dosages (0, 200, 300, 400, 500, 1000 nM). Following BRD3308 treatment the cells were incubated for 36 h and the mean fluorescent intensity of Gr1 (APC) was assessed using flow cytometry.

**RNA sequencing of ER-Hoxb8 cells following treatment with BRD3308.** ER-Hoxb8 cells stably expressing ERG variants and an empty vector for control were carefully washed to remove beta-estradiol followed by treatment with either 5 μM BRD3308 or DMSO. We conducted four independent experiments for each sample type. The cells were incubated for 36 h fluidized in TRIzol reagent and prepared for RNA sequencing as described above. Data analysis was conducted as described for murine fetal liver-derived HSPC.

**RGFP966 treatment in B-ALL cell lines.** Cell lines were seeded in a 24-well plate (150,000 cells/well, 0.5 mL/well RPMI 10%FBS) and treated with DMSO or RGFP966. After 48 h, each condition was harvested and resuspended in 190 μL PBS to which 5 μl Counting Beads (Spherotech AccuCount Blank Particles ACBP-70-10) and 5 μl 7-AAD (eBioscience™ 7-AAD Viability Staining Solution) were added. Measurement was acquired on BD LSRFortessa™. Data was analyzed with FlowJo software (BD). Live-cell (7-AAD) count was normalized to bead count.

**Inhibition of HDAC3 in vitro in AML lines and prostate cancer cells.** In a 24-well plate, 50k leukemia and prostate cancer cells were seeded per well. Cells were treated with either DMSO (control) or HDAC3 inhibitor (RGFP966) at a final concentration of 1 μM. After 48 h, leukemia cells were harvested for flow cytometric counting, and 7-AAD was used to distinguish live from dead cells. On the 7th day following the inhibition of HDAC3, prostate cancer cells were harvested for flow cytometry analysis, and 7-AAD was used to distinguish between live and dead cells.

### CRISPR-Cas9-mediated targeting of ERG in leukemic cell lines
**Cloning and virus preparation of single guide RNA.** Single guide RNA (sgRNA) against human *ERG* was cloned into CRISPR V2 plasmid

(addgene, #52961). Three million HEK-293T cells were seeded per plate of 100 mm dish. A total 10 dishes were used for per guide of human ERG. Three sgRNAs were chosen (see below): two targeting exon 2 while one target exon 5[24]. The transfection of 293T cells was performed using calcium chloride transfection based on the manufacturer's instructions. The virus was collected twice after 48 and 78 h. The virus was then filtered through a 0.2 μM filter. To achieve the final concentration of 10%, 10 mL of PEG8000 (40%) was added to 30 mL of virus. After being added to PEG8000, the virus was kept overnight at −20 °C. The next day, the virus was thawed at room temperature and centrifuged for 15 min at 4000 rpm using a swigging bucket centrifuge. After discarding the supernatant, the virus (30 mL) was resuspended in 200 L, aliquoted, and stored at −80 °C for future use.

| *ERG* targeting guides | Sequence | Position | PAM |
|---|---|---|---|
| Guide1 | GTGGGCAGCCCAGACACCGT | Exon 5 | TGG |
| Guide2 | GTCCTCACTCACAACTGATA | Exon 2 | AGG |
| Guide3 | AGCCTTATCAGTTGTGAGTG | Exon 2 | AGG |

**KO of *ERG* in leukemic cell lines.** Each well of a 12-well plate was seeded with 1 million cells and polybrene was added at a final concentration of 6 μg/mL. Cells were kept at 37 °C in the presence of 5% $Co_2$ for 30 min. Thereafter, the *ERG*-targeting Guide RNA 1, 2, and 3 were pooled with a non-targeting Guide virus. A total of 100 μL of pooled Erg virus and Non-targeting guide RNA virus was added to the cells. Spinfection was performed by spinning plates at a speed of $1000 \times g$ at 32 °C for 90 min. The cells were kept at 37 °C with 5% carbon dioxide after spinfection. Approximately 24 h after transduction, puromycin was added at a final concentration of 1 μg/mL.

**Apoptosis analysis.** Cells were seeded at 20,000 per well of a 48-well plate after 2 days of puromycin selection. Cells were harvested for annexin-7AA staining every 4 days. Harvested cells were centrifuged at $300 \times g$ for 5 min at 4 °C, then washed with PBS. After discarding the supernatant, cells were stained with 100 μL Annexin-APC antibody diluted in Annexin buffer (1:20 dilution). Cells were kept in darkness at room temperature for 30 min. The cells were then washed with PBS and centrifuged. The supernatant was discarded, and the cells were suspended in staining media (PBS + 2% FBS) with 7AAD (a dilution of 1:100). Flow cytometry was used to analyze the cells after 5 min incubation.

### HDAC3 inhibition and CRISPR targeting of HDAC3 in vivo
**HDAC3 inhibition using RGF966 in NSG mice.** Busulfan (20 mg/kg) was administered intravenously to NSG mice 24 h prior to transplantation. SKNO1 cells (250 K) or ELF-153 cells (10⁶) were transplanted through the tail veins of 24 NSG mice (vehicle vs. RFGP966). One day following transplantation, mice were treated intraperitoneally with either 15 mg/kg of RGFP966 or vehicle once daily for 21 days. The mice were then sacrificed, and the bone marrow was analyzed using flow cytometry for human CD45 to assess disease burden.

RGFP966 (Selleck) was dissolved in DMSO (final concentration−7%), PEG300 (40%), Tween 80 (10%) and 0.9% w/v NaCl (43%). A similar concentration of ingredients was used for the vehicle.

**Targeting HDAC3 using CRISPR-dCas9 system in SKNO1 cells.** CRISPR-dCas9 system was used to inhibit HDAC3 expression in SKNO1 cells. Monoclonal SKNO1 dCas9 cells were generated by transduction with a dCas9 lentivirus. Transduced cells were selected by adding blasticidin at 1 mg/mL final concentration. Single cells were seeded per well of a 96-well plate. To test the efficiency of clones, we transduced single cells with RFP virus which contained a guide RNA (sgRNA) that targeted the RFP in the original vector itself. In flow cytometry, selected clones exhibited an 85−89% reduction in RFP MFI

(mean fluorescent intensity). Selected SKNO1 dCas9 monoclonal cells were later transduced with lentivirus containing sgRNA targeting the transcription start site of the HDAC3 gene. Transduced SKNO1 dCas9 cells were selected by adding puromycin to the media at a final concentration of 1 μg/mL. Cells were maintained in puromycin for 14 days, and the expression of HDAC3 was measured at the RNA and protein levels (Fig. S8).

Once HDAC3 inhibition was confirmed, 250k sgNT-SKNO1 (non-targeting) dCas9 cells and sgHDAC3-SKNO1 dCas9 cells were transplanted per mouse. Mice were sacrificed 45 days after transplantation. The bone marrow cells were harvested and stained with an anti-hCD45 antibody to determine the disease burden. The same procedure was conducted for targeting HDAC3 in THP1 cells.

| *HDAC3* targeting guides | Sequence |
|---|---|
| Guide1 | CGGCACCATGGCCAAGACCG |
| Guide2 | GTAGAAATAGGCCACGGTCT |
| Guide3 | ATAGGCCACGGTCTTGGCCA |

## Reporting summary

Further information on research design is available in the Nature Portfolio Reporting Summary linked to this article.

## Data availability

Data generated during the study are available in a public repository. Data generated during the study are available in a public repository. BioID−Repository: available at massIVE (https://massive.ucsd.edu) with access code ID MSV00089158 and in ProteomeXchange with access code ID PXD032874. ChIP sequencing−GEO accession number: GSE200389. Link− https://www.ncbi.nlm.nih.gov/geo/query/acc.cgi?acc=GSE200389 RNA sequencing (ER-Hoxb8 cells): GEO accession number: GSE200391. Link− https://www.ncbi.nlm.nih.gov/geo/query/acc.cgi?acc=GSE200391 RNA sequencing (Fetal liver-derived HSPC): GEO accession number: GSE200392. Link− https://www.ncbi.nlm.nih.gov/geo/query/acc.cgi?acc=GSE200392 The data is all publicly available. Release date−May, 26, 2023. Source data files and uncropped Western blot images from this study are provided with this paper. Source data are provided in this paper.

## Code availability

The code generated during the current study is available from the corresponding author on reasonable request and without restrictions. The analyses of RNA-seq and ChIP-seq data in this study were conducted using standard and published algorithms using R software, as described in the Methods section. Any further details or clarifications regarding the code used can be obtained from the corresponding author upon request.

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

## Acknowledgements

The authors thank Jonatan Barel, Gilgi Friedlander, Dayana Yahalomi, and Avital Sarusi Portuguez from the Nancy and Stephen Grand Israel National Center for Personalized Medicine, Weizmann Institute of Science, for RNA-seq and ChIP-seq analysis service. The authors thank Brice Mouttet and the Bourquin lab for assistance in finalizing Co-IP protocols. We thank Ercument Dirice for sharing substantial information regarding BRD3308 handling in mice. The authors would also like to thank Rani Elkon for providing general bioinformatics guidance. We are indebted to Itzhak Ben Moshe (Ofer) for help in mice processing and in vivo experiments. We thank all past and present members of the S.I. research group for fruitful discussions and advice. This study was supported by a CIHR grant to GGP, the Israel Science Foundation, Dotan Research Center in Hemato-Oncology to SI, the Waxman Cancer Research Foundation to GGP and SI, CRISPR-IL consortium, the Israel Cancer Research Foundation professorship grant, the Israel Children's Cancer Foundation, the Fight Kids Cancer and the Nevzlin Foundations to SI. SI is the Gregorio and Dora Shapiro Endowed professor for Hematological Malignancies at Tel-Aviv University.

## Author contributions

Conceptualization and methodology, S.I., E.K., Y.B., and S.M.; Formal analysis, E.K., S.M., and D.Y.; Investigation, E.K., S.M., D.Y., J.I.T., J.S., J.Z., B.R., A.R., Y.B., H.F., and I.G.; Resources, D.B.S., J.S., J.Z., Y.Z., A.D., V.P., M.Y., N.A., C.C.W., A.R., G.G.P., and M.M.; Writing—original draft, E.K., S.I.; Supervision, S.I., J.P., G.G.P., and M.M.

## Competing interests

The authors declare no competing interests.
