## [Peer Review File · Nature Communications]

REVIEWER COMMENTS

Reviewer #1 (Remarks to the Author):

The manuscript by Kugler et al. reported the novel finding that Pro199 is required for ERG-dependent leukemia development. P199L mutation affects several oncogenic properties of ERG. The mechanisms were studied. P199L mutation was found to impair the repression control of ERG, and abrogate the ability of ERG to block myeloid differentiation. The authors also obtained evidence that ERG interacts with the N-CoR/HDAC3 complex and that P199L impairs this interaction. Finally, chemical and genetic inhibition of HDAC3 affected ERG-dependent human leukemia cell progression. These results are novel and potentially important, suggesting that ERG-mediated repression via the NCoR-HDAC3 complex plays an important role in leukemia progression. However, several issues need to be addressed to strengthen the conclusion regarding the involvement of the N-CoR/HDAC3 complex.

1. The current results have not demonstrated a direct interaction between ERG and the N-CoR/HDAC3 complex. BioID is a proximity assay. In addition, the study was performed in HEK293 cells under ectopic expression conditions. Please validate the interaction in AML cells by Co-IP.
2. An alternative explanation is that P199L interacts more strongly with HATs (such as p300 which catalyzes H3K27ac). To exclude this possibility, it is important to perform ChIP-Seq on p300 and NCoR/HDAC3 in ERG and ERG-P199L cells. If the hypothesis is correct, p300 levels should be similar in both cells, while the levels of NCoR/HDAC3 should be reduced in P199L cells.
3. Please explain why H3K9 levels were not affected. Given that HDAC3 deacetylates both H3K27 and H3K9, a reduction of ERG-HDAC3 interaction should increase both H3K27ac and H3K9ac.
4. In Fig. 8, the authors show that SKNO-1 and TF1 are both ERG and HDAC3 dependent. However, the results have not demonstrated an interaction between ERG and HDAC3. To do so, the authors need to include sgERG+/-shHDAC3. Have the authors tried to engraft sgERG-SKNO-1 or sgERG-TF1 cells? Alternatively, they may show that downregulation of HDAC3 diminishes the difference between WT and P199L ERG in transgenic models (Fig. 3A).
5. In Fig. 2A, can the authors also show the changes of specific CFUs? I would predict that myeloid CFUs (e.g., CFU-GM) should be higher in P199L, as P199L increases myeloid differentiation (Fig. 2F).

Reviewer #2 (Remarks to the Author):

In this study, Eitan Kugler and colleagues have established from a transgenic mouse model which was initially intended to be used to characterize the effect of ERG overexpression, that a critical proline site located at the N-terminal region of the PNT domain of ERG is essential to promote ERG-mediated leukemic transformation. The authors further demonstrated that this site is indeed required for the interaction between ERG and the NCoR-HDAC3 complex to block hematopoietic stem cell differentiation. Consistent with this observation, the inhibition of HDAC3 alters viability of ERG-dependent AML cell lines both in vitro and in vivo. This study is technically sound, well written, and brings new insight into the mechanisms by which ERG dysregulation contributes to leukemia transformation. Specific concerns will have to be addressed to strengthen the findings and support the claim that ERG directly interacts with the NCoR-HDAC3 complex.

Specific concerns:

1) Related to Figure 3, the authors showed nicely that the overall survival of mice overexpressing the mutant form of ERG (P199L-ERG) is significantly enhanced in comparison with animals expressing the WT form of ERG, thereby suggesting that the proline 199 supports most of the oncogenic activity of ERG. Given that these experiments were carried out upon overexpression of various ERG constructs and that ERG proteins form homo and heterodimeric complexes, we cannot exclude an effect of the endogenous ERG in these observed effects. Could the authors assess the impact of a non-targetable version of P199L-ERG on mouse survival in the context of endogenous ERG suppression using shRNAs? In addition, does the overexpression of a mutant form of ERG deleted from the PNT domain phenocopy the effects observed upon overexpression of P199L-ERG both in term of overall mouse survival and immunophenotype of the leukemia that may eventually arise?

2) In line with this question, did the authors retrieve P199L-ERG-positive AML cells from the 37.5% of mice who developed a disease (Figure 3A) to investigate the mechanism by which AML developed despite the absence of functional ERG. Was this still dependent on endogenous ERG? Was this leukemia progression independent on ERG? What was the immunophenotype of these leukemic cells in comparison with those of animals which were injected with HSPCs infected with WT ERG?

3) In their RNAseq data from Figure 4A, did the authors notice the activation of specific transcriptional programs related to megakaryocytic differentiation upon P199L-ERG and WT ERG overexpression that could be consistent with the results that they have obtained in Figure 2C-F?

4) Related to Figure 5, could the author confirm their interactome data using co-immunoprecipitation experiments between WT-ERG, P199L-ERG and the NCoR-HDAC3 complex? What is the primary interactor of ERG amongst the proteins of this complex? Is this interaction direct? Does the deletion of the PNT abrogate the interaction between ERG and its primary interactor by co-IP experiments? Does the overexpression of P199L-ERG affect the binding of endogenous ERG to the NCoR-HDAC3 complex? Finally, does the inhibition of HDAC3 affect the interaction between ERG and the NCoR-HDAC3 complex?

5) From Figure 6, the authors showed that the overexpression of WT-ERG reduces chromatin activity at enhancer regions of genes involved in myeloid differentiation. By combining this observation with their transcriptomic data from Figure 4, is there a core set of restricted, top-scoring myeloid differentiation-related genes that are the most affected by WT-ERG overexpression? Is the expression of these genes altered by P199L-ERG and HDAC3 inhibitors? It would be great to depict the identity of these genes in a single heatmap cross comparing all these conditions.

6) Although we acknowledge the fact that the authors profile the dependency of various AML cell lines to ERG in order to pick up their most representative cell lines for the study of the effect of HDAC3 inhibition on their growth in vivo (Figure 8F and 8G), we cannot exclude the possibility that other confounding variables (i.e., the various genetic backgrounds of these cell lines) could account for their differential sensitivity to HDAC3 inhibition. To rule out such possibility, could the authors carry out an additional in vivo study investigating the effect of HDAC3 inhibition on the growth of their ER-Hoxb8 empty backbone/WT-ERG isogenic system and on the overall survival of mice injected with these isogenic cells?

7) What is the differentiation status of the ERG-dependent and -independent human AML cell lines treated with HDAC3 inhibitors both in vitro and in vivo?

Reviewer #3 (Remarks to the Author):

The study by Kugler et al demonstrates a novel interaction between ERG and the NCoR-HDAC3 co-repressor complex, which is required for its oncogenic activity. They discovered a critical role for the conserved amino-acid proline at position 199 at the 3' end of the PNT domain. This they suggest is crucial for ERG's leukemogenic activity. Furthermore, they suggest that modulation of this interaction may provide an opportunity for therapeutic intervention. This is a very well-designed, executed, and written study. The findings are innovative and potentially have clinical implications.

1) The HDAC and SWI/SNF connections were observed in the proximity assays. Is there any role for the polycomb 2 complex? EZH2 inhibitors have been considered for certain leukemias. HDAC inhibitors may also act to inhibit the methylation effects of EZH2.

2) Other ERG-related cancers are mentioned. With the findings in this study have any relevance to non-myeloid tumors?

We were pleased to read the positive evaluation of our manuscript by the editor and reviewers. We found the reviewers' comments very helpful and conducted additional analyses and experiments to improve the manuscript per the suggestions of the reviewers.

While reading the constructive comments of the reviewers we have realized that the focus of the paper has been somewhat miss presented. The fortuitous finding of the markedly reduced leukemogenic activity of P199L ERG served just as a tool for the major discovery that the interaction of ERG with the HDAC3-NCOR complex is important for ERG's role in leukemia. Hence, we changed the title to reflect this discovery. In accordance with this change and as a response to the last comments of reviewer 3, we added new experiments demonstrating the general requirement of HDAC3 to ERG-driven myeloid and lymphoid leukemia as well as prostate cancer. They are described at the end of this document in our responses to reviewer 3. These additions come as significant new publications further substantiate the leukemogenic importance of ERG[1, 2]. Thus, thanks to the constructive reviewers our discovery is of broader relevance with a potential for clinical translation.

The following is a point-by-point response to these comments and a description of the improvements made to the manuscript as a result.

Reviewer #1

We appreciate the reviewer's evaluation as to the novelty and importance of our findings.

Comment1

The current results have not demonstrated a direct interaction between ERG and the N-CoR/HDAC3 complex..... Please validate the interaction in AML cells by Co-IP.

Response:

Co-IP was performed in ERG dependent AML cell lines (SKNO1, TF1, figure Rev1) to validate the interaction between ERG and the NCoR/HDAC3 complex. Moreover, the interaction was also validated in VCaP cells (prostate cancer cells harboring the TMPRSS-ERG translocation) as also suggested by reviewer #3, comment 2 (figure Rev16).

We demonstrated by Co-IP that ERG interacts with NCOR2/SMRT, the main scaffolding structure for NCOR-HDAC3. As a result of its small size, it is quite challenging to pull down HDAC3, and we are not certain whether ERG interacts directly with HDAC3 or indirectly via NCOR2.

Changes in manuscript – Co-IP results were added as figure 8A, and the text in the manuscript was modified accordingly – lines 352-354.

Figure Rev1:

Figure Rev1 - Co-IP assays for endogenous ERG and the NCoR/HDAC3 complex in ERG dependent AMLs. IP for endogenous ERG and blot for NCoR2 and HDAC3 in SKNO1, TF1 and ELF-153 cells. Representative figure of three independent experiments is shown.

Comment 2:

An alternative explanation is that P199L interacts more strongly with HATs (such as p300 which catalyzes H3K27ac).

Response:

We thank the reviewer for this comment and the alternative hypothesis. Having a similar hypothesis. as part of our analysis of the Bio-ID results, we specifically examined the interaction between ERG variants and P300. The distribution of spectral counts between ERG variants did not differ statistically, and there was a trend for **decreased** proximity between P199L-ERG and P300.

Bio-ID run	WT-ERG	P199L-ERG	Control
1	22	0	0
2	63	31	0
3	40	17	0
4	9	15	0
p.value=0.15			

Table Rev1. Normalized spectral counts for P300 across ERG variants. P value was calculated using two tailed t-test.

Comment 3:

Please explain why H3K9 levels were not affected. Given that HDAC3 deacetylates both H3K27 and H3K9, a reduction of ERG-HDAC3 interaction should increase both H3K27ac and H3K9ac.

Our findings demonstrate that P199L partially disrupts the interaction of the NCoR-HDAC3 complex with ERG. In ERG-expressing ER-Hoxb8 cells, H3K27 deacetylation was observed primarily on genomic sites overlapping H3K27ac-H3K4me1 but not H3K27ac-H3K9ac (figure 6E). These sites were related to genes expressed in mature myeloid cells. Therefore, we concluded that ERG-NCoR-HDAC3 regulate enhancers and not promoters of myeloid target genes, thus affecting mainly H3K27ac and not H3K9ac.

Comment 4:

In Fig. 8, the authors show that SKNO-1 and TF1 are both ERG and HDAC3-dependent. However, the results have not demonstrated an interaction between ERG and HDAC3. To do so, the authors need to include sgERG+/-shHDAC3. Have the authors tried to engraft sgERG-SKNO-1 or sgERG-TF1 cells? Alternatively, they may show that downregulation of HDAC3 diminishes the difference between WT and P199L ERG in transgenic models.

In both SKNO-1 and TF-1, a knockout of ERG results in cell death. Consequently, this experiment cannot be conducted. In addition, unfortunately, the P199L transgenic ERG mice were eliminated upon their discovery 10 years ago. Only much later had we realized that this fortuitous “accident” was an opportunity for discovering the mechanism of leukemogenesis by ERG.

Comment 5:

In Fig. 2A, can the authors also show the changes of specific CFUs? I would predict that myeloid CFUs (e.g., CFU-GM) should be higher in P199L, as P199L increases myeloid differentiation (Fig.2F).

We do not have data on specific colonies based on their morphology. Flow cytometry was performed on the total bulk of cells collected at each round of re-plating. In support with the reviewer’s hypothesis, a significant increase in the percentage of c-Kit/Sca1 double positive HSPCs was demonstrated in cells transduced with WT-ERG, accompanied by a significant decrease in the expression of mature myeloid markers (figure 2E-F). Thus WT-ERG but not P199L ERG markedly restricts myeloid differentiation of transduced HSPCs.

Reviewer #2

We thank the reviewer for the positive evaluation regarding the importance of our study for the understanding of the leukemogenesis mechanism by ERG.

Comment 1:

What is the effect of P199L-ERG on endogenous ERG ? Could the authors assess the impact of a non-targetable version of P199L-ERG on mouse survival in the context of endogenous ERG suppression using shRNAs? In addition, does the overexpression of a mutant form of ERG deleted from the PNT domain phenocopy the effects observed upon overexpression of P199L-ERG ?

We would like to thank the reviewer for this important comment. It has been shown that ERG does not form homodimers, unlike ETV6[3, 4], but the reviewer's question regarding the potential effect of exogenous expression of WT or mutant ERG on endogenous ERG is both interesting and important.

Since silencing endogenous *Erg* in primary murine fetal liver progenitor cells is lethal, as previously reported [5, 6], complementary assays were required to determine whether P199L-ERG had a dominant negative effect over endogenous ERG.

To answer this question, we first need to emphasize that endogenous ERG in murine fetal liver derived HSPC have no direct role in leukemia progression since HSPC transduced with an empty vector and transplanted to immunocompetent mice do not develop leukemia (figure 3A). Only the exogenous expression of ERG transforms HSPC and promotes leukemogenesis. This probably implies that only high levels of ERG are leukemogenic. Therefore, even if P199L-ERG exerts a dominant negative effect on endogenous ERG, it does not explain the absence of leukemia development.

In addition, P199L is not a complete loss of function mutation. HSPC derived from murine fetal liver exhibit a distinct phenotype following transduction of P199L-ERG in comparison with HSPC transduced with an empty vector. P199L-ERG transduced HSPC express stem cell surface markers (cKit, Sca1) and megakaryocyte markers (CD41, CD61) similarly to WT-ERG transduced HSPC (figures 2A-D). As we have initially reported in 2005, ERG has, in addition to its role in hematopoietic normal and malignant stem cell biology, a positive role in megakaryocytic differentiation[7, 8]. Accordingly, the forced expression of both WT and mutated ERG in K562 cells (erythroleukemia with low endogenous ERG expression) induces an erythroid to megakaryocytic phenotypic switch (figure Rev2A). This is also confirmed in a reporter assay with a megakaryocytic promoter of CD41 conducted in 293T cells (figure Rev2B). Moreover, as discussed in the manuscript and shown in figure 2F, forced expression of P199L-ERG restricted myeloid differentiation as compared to the control in murine fetal liver-derived HSPC, yet not to the same degree as the WT-ERG. Together, these findings suggest that P199L-ERG is functional, but not at the same extent as WT-ERG. Therefore, a dominant negative effect of mutated ERG cannot fully explain the observed phenotype in K562 cells and fetal liver HSPC upon force expression of P199L-ERG.

To further explore the potential functional interactions of mutated P199L and endogenous ERG we utilized the ERG+85 stem cell enhancer reporter system as we previously described[9, 10]. The ERG +85 stem cell enhancer contains binding motifs for multiple transcription factors, including ERG itself (heptad of transcription factors - SCL/TAL1, LMO2, LYL1, ERG, FLI1, RUNX1, GATA2). The ERG + 85

region has super-enhancer characteristics in human HSPCs and in a large subset of human AMLs and correlates with a high expression of several stemness regulators. In 66 human AML samples, the transcript that exhibited the highest positive correlation with the intensity of the ERG enhancer H3K27 acetylation was found to be *ERG* mRNA[10]. We hypothesized therefore, that a dominant negative effect of P199L-ERG may negatively affect transactivation of the ERG +85 enhancer and disrupt endogenous *Erg* expression in murine HSPCs.

RNA sequencing of murine fetal liver-derived HSPCs and ER-Hoxb8 cells following transduction with ERG variants does not support this hypothesis, demonstrating that WT and mutated human ERG samples display no significant difference in endogenous mouse *Erg* expression (figure Rev3). Cells transduced with an empty vector (BB) exhibit relatively low levels of endogenous *Erg* expression, likely due to more mature differentiation state of these cells compared to those expressing ERG variants[5] (table Rev2). As a next step, we applied the ERG +85 enhancer luciferase reporter assay to 293T cells. Force expression of P199L-ERG reduced transactivation of the enhancer compared to WT-ERG (figure Rev4A). However, the co-expression of WT and P199L-ERG activated the enhancer similarly to the expression of WT-ERG alone suggesting against a negative dominant effect of P199L-ERG (figure Rev4B).

Lastly, we tested the ERG +85 system in ERG dependent human AML cells. P199L-ERG was forced expressed in SKNO1 cells stably expressing a lentiviral reporter system in which BFP expression is driven by ERG 85+ as we previously described [10]. Therefore, a decrease in transactivation of the enhancer would result in a decrease in BFP expression. The expression of P199L-ERG did not significantly reduce BFP signal nor decreased cell proliferation (CRISPR-Cas9 targeting of ERG in SKNO1 cells results in a decrease in cell proliferation, figure 8B), suggesting that P199L-ERG had no dominant negative effect over endogenous ERG (figure Rev5).

In reference to the second part of the comment, the phenotype of P199L-ERG is not equivalent to the phenotype of ERG with a deleted PNT domain (delPNT-ERG). Murine HSPC expressing delPNT-ERG retain self-renewal ability in re-plating assays and colony-forming unit assays and progress to AML upon transplantation to immune deficient mice (figure Rev6). The immune phenotype is similar to WT and P199L-driven ERG AML (CD34 negative, CD150 positive blasts). Interestingly, the morphology of delPNT-ERG colonies in semi-solid conditions differs from that of WT and P199L-ERG colonies. Colonies are more widely dispersed, and tend to be more proliferative (figure Rev6). These phenotype differences may be related to the major changes in ERG structure following deletion of the PNT domain as compared with P199L-ERG where no effect on ERG general structure was noted (figure 1 C-E), and may also suggest a complex role for the domain as a whole.

Changes in the manuscript:

In the chapter related to the effect of P199L-ERG on lineage commitment we added a paragraph describing the experiments conducted in K562 and luciferase reporter assays demonstrating the similar functionality of WT and P199L-ERG in features related to megakaryopoiesis (lines 149-154). Additionally, figure Rev2 is now displayed as supplementary figure S4.

In the same section, we have also added a paragraph that describes the phenotype of delPNT-ERG expression in HSPC both in vitro and in vivo, along with supplementary figures (line 167, figure S5).

Additionally, we discussed the potential effects of P199L on endogenous ERG in several sections of the revised manuscript. We have added an additional paragraph to the text and figure Rev4 and figure Rev5 as a supplementary figures (ERG + 85 enhancer reporter assay in 293T and SKNO1 cells) under the section entitled "P199L decreases the leukemic transformation of HSPC induced by ERG expression" (line 185, figures S6 and S7). In the section titled "P199L compromises ERG transcriptional repression signature in HSPC", we presented data on endogenous Erg expression in murine HSPC (figure Rev3 and figure S8A in the manuscript, line 242). We include data on endogenous Erg expression of Hoxb8 cells in the section titled "HDAC3 inhibition alleviates the myeloid differentiation block induced by ERG expression in hematopoietic progenitors" (line 321, figure S8B).

In the discussion section, we addressed in detail the issues of P199L functionality, the effect on endogenous ERG, and the discrepancies between P199L and delPNT-ERG phenotypes.

Figure Rev2. P199L-ERG is not a complete loss of function mutation. A. Force expression of ERG variants in K562 cells. n=3, CD61 expression was measured using flow cytometry. **B.** Activity of the CD41 promoter upon expression of ERG variants in 293T cells. Fold change of luciferase expression was measured. n=3, p=0.02, t test. . * p<0.05, ** p<0.01, *** p<0.001

Figure Rev3. Expression of endogenous *Erg* is similar in murine HSPC following WT and P199L-ERG force expression. Left panel – expression counts obtained from RNA sequencing of ER-Hoxb8 cells expressing ERG variants. Right panel – expression counts obtained from RNA sequencing of fetal liver derived HSPC cells expressing ERG variants. Normalized counts are presented.

Differentiation		ENSMUSG00000015053.11	ENSMUSG00000040732.15	ENSMUSG00000030786.15	ENSMUSG00000057666.15
Hours	Replicate	GATA2	ERG	ITGAM(CD11b)	GAPDH
0	A	3023	3632	176	65672
0	B	3368	4106	236	75793
4	A	5199	4012	159	111028
4	B	4283	3681	172	93035
8	A	7487	3615	286	86209
8	B	6703	3465	250	77798
12	A	5916	3487	700	72325
12	B	5358	3299	596	80049
24	A	3857	3018	1353	74852
24	B	4264	3219	1534	92318
36	A	2798	3195	5671	75378
36	B	3235	2935	5143	93095
48	A	2362	3191	19213	74978
48	B	2498	3361	15612	70729
72	A	1296	2572	62135	46954
72	B	1382	2469	48072	52041
96	A	514	1999	106734	40107
96	B	569	2445	104976	41652
120	A	310	651	97810	45616
120	B	325	459	63033	36412

Table Rev2. Normalized read counts for mouse Gata2, Erg, Gapdh, and CD11b during the 5-day differentiation time course of ER-Hoxb8 cells. RNA-seq. Courtesy of Dr. David Sykes.

Figure Rev4. Activation of the ERG +85 stem cell enhancer in 293T following ERG variants force expression does not support a dominant negative role for P199L-ERG. A. ERG +85 enhancer luciferase activity following transfection with ERG variants. n=3, ANOVA. B. ERG +85 enhancer luciferase activity following co-transfection of WT-ERG and P199L-ERG (ERG-Both). 500 ng each construct, n=4, ANOVA.

SKNO1-ERG+85-BFP-stem cell enhancer

B

*

C

Figure Rev5. Activation of ERG +85 stem cell enhancer reporter in ERG dependent SKNO1 cells following forced expression of P199L-ERG. A. Upper line - GFP marks transduced cells stably expressing the reporter. Bottom line - BFP marks the activation of the ERG +85 enhancer. **B.** summary of three independent experiments. $p=0.016$, t test. **C.** Western blot in SKNO1 cells transduced with P199L-ERG and backbone for control.

A

B

Figure Rev6. The phenotype of delPNT-ERG and P199L-ERG in transduction-transplantation assays in murine HSPC differ **A.** Re-plating assays in semi solid conditions of fetal liver derived murine HSPC transduced with ERG variants. CFU are presented. Media is supplemented with SCF, FLT3, TPO. **B.** Survival analysis of irradiated C57B/6 mice transplanted with murine fetal liver derived HSPC following transduction with ERG variants (10^5 transduced cells/mouse). A log-rank test was used to compare survival distribution between groups. * $p < 0.05$, ** $p < 0.01$, *** $p < 0.001$

Comment 2:

In line with this question, did the authors retrieve P199L-ERG-positive AML cells from the 37.5% of mice who developed a disease (Figure 3A) to investigate the mechanism by which AML developed despite the absence of functional ERG. Was this still dependent on endogenous ERG? Was this leukemia progression independent on ERG? What was the immunophenotype of these leukemic cells in comparison with those of animals which were injected with HSPCs infected with WT ERG?

As shown in figure 3C, the immune phenotype of P199L-ERG is similar to that of WT-ERG mice driven AML (CD34-, CD150 +). We believe that the endogenous Erg has no role in leukemia progression in the transduction-transplantation assays in C57B/6 mice. As was mentioned, HSPC transduced with an empty vector and transplanted to immunocompetent mice do not results in leukemia development. The P199L-ERG AML cells (37.5% of mice) were not analysed further. Please see also our detailed response to Reviewer 2 above and many experiments we added to rule out a potential functional interaction between P199L ERG and the endogenous ERG.

Comment 3:

In their RNAseq data from Figure 4A, did the authors notice the activation of specific transcriptional programs related to megakaryocytic differentiation upon P199L-ERG and WT ERG overexpression?

We thank the reviewer for this important comment. Both WT-ERG and P199L-ERG expression in murine HSPC resulted in upregulated megakaryocytic transcriptional program. The heatmap below illustrates the most significant differentially expressed genes affected upon ERG variants expression, and is consistent with the results shown in figure 2D. This is consistent with our initial report in 2005, that ERG has a role in megakaryocytic differentiation[7]. As we detail in our response to reviewer 2, in a series of experiments P199L ERG retains this megakaryocytic effect.

Figure Rev7. P199L and WT-ERG induce a megakaryocytic transcriptional program in murine HSPC. Fetal liver derived HSPC cells transduced with ERG variants. RNA was collected for sequencing after 72 hours in culture (SCF, TPO, FLT3, serum free media). The genes denoted are those enriched in the GSEA set “GO_PLATELET_ALPHA_GRANULE” (most significant set related to megakaryocytic differentiation, GO). Differentially expressed genes are presented, padj<0.05

Comment 4:

Related to Figure 5, could the author confirm their interactome data using co-immunoprecipitation experiments between WT-ERG, P199L-ERG and the NCoR-HDAC3 complex? What is the primary interactor of ERG amongst the proteins of this complex? Is this interaction direct? Does the deletion of the PNT abrogate the interaction between ERG and its primary interactor by co-IP experiments? Does the overexpression of P199L-ERG affect the binding of endogenous ERG to the NCoR-HDAC3 complex?

In 293T cells, we confirmed that WT-ERG, P199L-ERG, and delPNT-ERG interact with the NCoR-HDAC3 complex. All ERG constructs were pulled down with the NCoR-HDAC3 complex without noticing any significant differences (figure Rev8). These results do not contradict the decreased proximity between P199L-ERG and the repression complex observed in the Bio-ID assay, given that pull down is not sensitive enough to detect subtle changes in protein proximity. Additionally, the interaction between endogenous ERG and NCoR-HDAC3 was confirmed in human ERG-dependent AML cell lines (figure Rev1, figure 8A in the paper) and in prostate cancer cells (VCaP, figure Rev16, Figure S15A). The Co-IP assays suggest that the ERG-NCoR-HDAC3 interaction is direct. Yet, as shown in Figure 5B, spectral counts for the NCoR proteins were much higher than for other members of the complex (HDAC3, TBL1XR1, GPS2), indicating a more direct interaction with the former and indirect interaction for the latter.

Figure Rev8. Co-IP assays for ERG variants and the NCoR/HDAC3 complex in 293T cells. Stable HEK293 line with inducible overexpression of SMRT/TBL1XR1/HDA3/GPS2 was generated. Purification of tagged SMRT was conducted using GFP binding resin.

Comment 5:

Is there a core set of restricted, top-scoring myeloid differentiation-related genes that are the most affected by WT-ERG overexpression? Is the expression of these genes altered by P199L-ERG and HDAC3 inhibitors?

We would like to thank the reviewer for this helpful comment that has been now incorporated into the manuscript. Figure Rev9 (figure 7G in the revised manuscript) displays genes associated with myeloid differentiation that are repressed following expression of WT-ERG in ER-Hoxb8 cells. Differentially expressed genes are presented ($\text{padj} < 0.05$). These genes are also repressed by P199L-ERG, although to a lesser extent, supporting the hypothesis that P199L interferes with myeloid restriction. Moreover, there is a subtle difference in the magnitude of gene repression between WT and mutated ERG, suggesting that P199L is not a complete loss of function mutation. Inhibiting HDAC3 results in the de-repression of these genes. Taken together, the gene expression pattern and the cell phenotype (figure 2F) are in agreement with a decrease in proximity to the NCoR-HDAC3 repression complex associated with P199L-ERG rather than a complete disruption of the interaction.

Figure Rev9. The effect of HDAC3 inhibition on myeloid differentiation related genes in ER-Hoxb8 cells expressing ERG variants. Gene expression matrix is presented as heatmap for differentially expressed genes ($\text{padj} < 0.05$)

Changes in the manuscript:

In the section “HDAC3 inhibition alleviates the myeloid differentiation block induced by ERG expression in hematopoietic progenitors” we added a paragraph (line 335) discussing the pattern of expression of myeloid differentiation related genes as presented in figure Rev9 and detailed in the text above. Additionally, Figure Rev9 has been included as figure 7G in the manuscript.

Comment 6:

We cannot exclude the possibility that other confounding variables (i.e., the various genetic backgrounds of these cell lines) could account for their differential sensitivity to HDAC3 inhibition. To rule out such possibility, could the authors carry out an additional in vivo study investigating the effect of HDAC3 inhibition on the growth of their ER-Hoxb8 empty backbone/WT-ERG isogenic system and on the overall survival of mice injected with these isogenic cells?

Since ER-Hoxb8 cells are estrogen-dependent and differentiate rapidly to mature neutrophils in the absence of estrogen, it cannot be transplanted into immune-deficient mice. As an alternative approach, we transplanted ELF-153 (FAB-M7, hypodiploid karyotype) ERG dependent human AML cells into NSG mice and treated the mice with IP RGFP966 once daily for 21 days (15 mg/kg). RGFP966 treatment resulted in a significant lower bone marrow blast counts at the end of the treatment period, as shown in the figure Rev11. Furthermore, as we mention in the opening paragraph in response to all reviewers, we significantly expanded our observations and discovered that HDAC3 inhibition has a role in multiple ERG driven malignancies.

Changes in manuscript -

The observation in the ELF-153 in-vivo experiment has been added to the text and is shown in figure 8G (line 395).

Figure Rev11. Pharmacologic inhibition of HDAC3 results in reduced leukemia growth in-vivo. NSG mice were transplanted with ERG-dependent human AML cells (ELF-153, 10^6 cells per mouse). The percentage of blasts at bone marrow was measured using flow cytometry after 21 days of treatment with IP RGFP966 (15mg/kg) compared with DMSO. n=8 for RGFP966, n=7 for DMSO. p=0.0205, t-test.

Comment 7:

What is the differentiation status of the ERG-dependent and -independent human AML cell lines treated with HDAC3 inhibitors both in vitro and in vivo?

Considering the data we presented regarding the effects of ERG on myeloid differentiation genes, this comment is excellent. We therefore examined the effects of HDAC3 inhibition on the differentiation status of ERG-dependent AML cells by flow cytometry (Figure Rev12).

Following HDAC3 inhibition, the expression of hCD45 was significantly increased across cell lines, indicating a more mature cellular component. Conversely, the expression of CD34 and CD117 was elevated in TF1 and ELF-153 lines. SKNO1 cells (translocation 8;21, AML1-ETO rearrangement, CD3 -, CD4 +, CD13 +, CD14 -, CD15 -, CD19 -, CD33 +) exhibit an increase in MFI for myelo-monocytic differentiation markers, which was not statistically significant (figure Rev12A). Both TF1 (erythroleukemia, CD3 -, CD13 +, CD14 -, CD15 -, CD19 -, CD33 +, CD34 +, CD71 +, HLA-DR +) and ELF-153 (hypodiploid AML, CD3 -, CD4 +, CD13 +, CD14 -, CD15 -, CD19 -, CD33 +, CD34 +, HLA-DR +) displayed features of monocytic differentiation as demonstrated by an significant increase in CD14, CD36, CD64, and HLA-DR MFIs (figure Rev12B-C).

It is difficult to draw a definitive conclusion from our FACS data since aberrant expression of cell surface antigens is common in AML[11]. Moreover, the exact mechanism underlying the sensitivity of ERG-dependent human AML lines to HDAC3 inhibition is still unknown. Rescue from myeloid differentiation block, however, may have a mechanistic basis, as demonstrated for TF1 and ELF-153 cells.

Changes in manuscript – Immunophenotype results were added as supplemental figure S13 and is described in the paragraph beginning at line 369.

A

B

C

Figure Rev12. Flow cytometry analysis of ERG dependent human AML cell lines following pharmacologic HDAC3 inhibition. Immune phenotype was assessed for SKNO1, TF1 and ELF-153 cells following treatment with RGFP966 for 96 hours. n=3, * p<0.05, ** p<0.01, *** p<0.001, t-test.

Reviewer #3

We greatly appreciate the positive recognition of the reviewer on the potential therapeutic implications of our findings.

Comment 1:

The HDAC and SWI/SNF connections were observed in the proximity assays. Is there any role for the polycomb 2 complex? EZH2 inhibitors have been considered for certain leukemias. HDAC inhibitors may also act to inhibit the methylation effects of EZH2.

The Bio-ID results did not indicate an interaction between ERG and members of the PRC2 complex. As such an interaction has been demonstrated for Tmprss2-ERG in prostate cancer[12, 13], we treated ERG dependent and independent human AML cell lines with EZH2 and SUZ12 inhibitors. Across all groups, no significant differences in cell growth were observed (figure Rev13A). However, ERG-dependent AML cells did not proliferate from day six onward after CRISPR-Cas9 was used to target EZH2 and SUZ12 (figure 13B). This result should be established and further explored in future studies.

Figure Rev13. The CRISPR-Cas9 targeting of EZH2 and SUZ12 inhibits the growth of ERG dependent AML cells. A. Pharmacological inhibition of EZH2. B. CRISPR-Cas9 targeting. n=3

Comment 2:

Other ERG-related cancers are mentioned. Are the findings in this study have any relevance to non-myeloid tumors?

Thank you for this very important point. We first added an in-vivo experiment with another ERG-dependent megakaryocytic AML line ELF-153 (figure Rev11). In addition, we included more ERG dependent AML cell lines (figure Rev14 and figure 8D-E in the revised manuscript). Schmoellerl et al. reported recently that EVI1-driven poor prognosis AMLs are ERG dependent[1]. Accordingly, figure Rev 14C demonstrates the exquisite sensitivity of HNT-34: RUNX1/EVI1 AML cells to HDAC3 inhibition (0.1 micromolar RGFP966 was sufficient to significantly decrease cell growth) compared with the resistance of MLL (KMT2A) rearranged AML cells. The manuscript was modified accordingly (lines 364, figure 8E).

ERG dependency is not limited to AML. According to the DepMap illustrated in figure Rev15A, numerous ALL cell lines are dependent on ERG. Specifically, in the poor prognosis B-ALL type harboring the TCF3-HLF translocation, ERG cooperates with TCF3-HLF to regulate enhancer functions and is essential for leukemia maintenance[14]. Moreover, a drug screen conducted previously on the HAL01 cell line (TCF3-HLF fusion) demonstrated high sensitivity to RGFP966 (figure Rev15B)[15]. We have therefore examined the effect of HDAC3 inhibition on the survival of patient-derived xenograft (PDX) of TCF3-HLF B-ALL as was previously described[16]. As shown in figure Rev15C, the PDX samples were sensitive to RGFP966 at concentrations comparable to those used for AML cell lines.

Furthermore, we assessed the effect of HDAC3 inhibition on prostate cancer cell survival. ERG plays a critical role in prostate cancer tumorigenesis[17]. In more than half of prostate cancer patients, the TMPRSS2-ERG translocation promotes the disease[18]. As a first step, Co-IP was used to confirm the interaction between ERG-NCoR-HDAC3 in VCaP cells (figure Rev16A). As shown in figure Rev16B, VCaP cells (TMPRSS2-ERG driven) were significantly more sensitive to HDAC3 inhibition as compared with LNCaP (low expression of endogenous ERG). Also note the relatively low expression of endogenous NCoR2-HDAC3 proteins in LNCaP cells (figure Rev16C).

These findings indicate a possible role for the ERG-NCoR-HDAC3 interaction in tumorigenesis also in non-myeloid malignancies, and suggest targeting HDAC3 as a possible therapeutic option.

Changes in manuscript - figures S14 and S15 have been added to the revised manuscript to illustrate the sensitivity of ALL and TCF3-HLF PDX cell lines, as well as prostate cancer cells. In addition, the text was modified accordingly in the abstract (line 51), the results sections (lines 377), and additionally in the discussion (line 465).

Figure Rev14. ERG dependent AML cell lines are sensitive to HDAC3 inhibition. A. Westren blot for endogenous ERG and HDAC3 in AML cell lines. B and C. ERG dependent and independent AML cell lines treated with RGFP966 for 96 hours and collected for viability analysis with flow cytometry, n=3, ANOVA.

A

B

C

Figure Rev15. TCF3-HLF ALL sensitivity to HDAC3 inhibition. **A.** DepMap data demonstrating ERG dependency in various leukemic cell lines. **B.** Selective drug sensitivity scores (sDSS) for HALO1 cell line retrieved from the FORALL portal (<https://proteomics.se/forall/>). **C.** TCF3-HLF PDX treated for 6 days with RGFP966 (ex-vivo).

Figure Rev16. ERG driven prostate cancer cells are sensitive to HDAC3 inhibition A. Co-IP for endogenous ERG and the NCoR-HDAC3 complex. B. HDAC3 inhibition using RGFP966 in VCaP (ERG dependent) and LNCaP (ERG independent) cells. Flow cytometry. $n=3$. C. WB for ERG and NCoR-HDAC3 proteins in VCaP and LNCaP cells. Red asterisk marks correct position of the band according to kDa estimation. ** $p<0.01$, t-test.

References:

1. Schmoellerl, J., et al., *EVI1 drives leukemogenesis through aberrant ERG activation*. Blood, 2022.
2. Kodgule, R., et al., *ETV6 Deficiency Unlocks ERG-Dependent Microsatellite Enhancers to Drive Aberrant Gene Activation in B-Lymphoblastic Leukemia*. Blood Cancer Discov, 2022.
3. Jousset, C., et al., *A domain of TEL conserved in a subset of ETS proteins defines a specific oligomerization interface essential to the mitogenic properties of the TEL-PDGFR beta oncoprotein*. EMBO J, 1997. **16**(1): p. 69-82.
4. Mackereth, C.D., et al., *Diversity in structure and function of the Ets family PNT domains*. J Mol Biol, 2004. **342**(4): p. 1249-64.
5. Knudsen, K.J., et al., *ERG promotes the maintenance of hematopoietic stem cells by restricting their differentiation*. Genes Dev, 2015. **29**(18): p. 1915-29.
6. Xie, Y., et al., *Reduced Erg Dosage Impairs Survival of Hematopoietic Stem and Progenitor Cells*. Stem Cells, 2017. **35**(7): p. 1773-1785.
7. Rainis, L., et al., *The proto-oncogene ERG in megakaryoblastic leukemias*. Cancer Res, 2005. **65**(17): p. 7596-602.
8. Salek-Ardakani, S., et al., *ERG is a megakaryocytic oncogene*. Cancer Res, 2009. **69**(11): p. 4665-73.
9. Aqaq, N., et al., *An ERG Enhancer-Based Reporter Identifies Leukemia Cells with Elevated Leukemogenic Potential Driven by ERG-USP9X Feed-Forward Regulation*. Cancer Res, 2019. **79**(15): p. 3862-3876.
10. Yassin, M., et al., *A novel method for detecting the cellular stemness state in normal and leukemic human hematopoietic cells can predict disease outcome and drug sensitivity*. Leukemia, 2019. **33**(8): p. 2061-2077.
11. Ossenkoppele, G.J., A.A. van de Loosdrecht, and G.J. Schuurhuis, *Review of the relevance of aberrant antigen expression by flow cytometry in myeloid neoplasms*. Br J Haematol, 2011. **153**(4): p. 421-36.
12. Kedage, V., et al., *Phosphorylation of the oncogenic transcription factor ERG in prostate cells dissociates polycomb repressive complex 2, allowing target gene activation*. J Biol Chem, 2017. **292**(42): p. 17225-17235.
13. Yu, J., et al., *An integrated network of androgen receptor, polycomb, and TMPRSS2-ERG gene fusions in prostate cancer progression*. Cancer Cell, 2010. **17**(5): p. 443-54.
14. Huang, Y., et al., *The Leukemogenic TCF3-HLF Complex Rewires Enhancers Driving Cellular Identity and Self-Renewal Conferring EP300 Vulnerability*. Cancer Cell, 2019. **36**(6): p. 630-644 e9.
15. Leo, I.R., et al., *Integrative multi-omics and drug response profiling of childhood acute lymphoblastic leukemia cell lines*. Nat Commun, 2022. **13**(1): p. 1691.
16. Frisimantas, V., et al., *Ex vivo drug response profiling detects recurrent sensitivity patterns in drug-resistant acute lymphoblastic leukemia*. Blood, 2017. **129**(11): p. e26-e37.
17. Lorenzin, F. and F. Demichelis, *Past, Current, and Future Strategies to Target ERG Fusion-Positive Prostate Cancer*. Cancers (Basel), 2022. **14**(5).
18. Tomlins, S.A., et al., *Antibody-based detection of ERG rearrangements in prostate core biopsies, including diagnostically challenging cases: ERG staining in prostate core biopsies*. Arch Pathol Lab Med, 2012. **136**(8): p. 935-46.

REVIEWER COMMENTS

Reviewer #1 (Remarks to the Author):

In this revised manuscript, the authors presented new data showing that endogenous ERG physically associated with the NCoR2/HDAC3 complex in several ERG-dependent cell lines by co-immunoprecipitation assays (Fig. 8A). One concern is that the results lack a specificity control (not pulled down by the anti-ERG antibody). In addition, can the authors provide details (including catalog#) of the antibodies used in these assays (I could not find this information for NCoR and HDAC3 antibodies).

Since the data have not demonstrated that ERG binds directly to NCoR2/HDAC3, can the authors discuss how ERG may physically interact with the NCoR/HDAC3 complex? As the authors pointed out, ERG is associated with the heptad complex (TAL1, LY11, LMO2, GATA2, RUNX1, ERG, FLI1). Is it possible that the ERG-NCoR2/HDAC3 interaction is mediated by some other components of the heptad complex?

The authors mentioned ERG-NCoR2/HDAC3 interaction throughout the manuscript. It wasn't entirely clear whether this refers to functional, physical (which may indirect) or direct interactions. However, the direct interaction has not been demonstrated.

Reviewer #2 (Remarks to the Author):

In this second version of the manuscript, the authors addressed satisfactorily all concerns raised by the reviewer. These responses contributed to enhance even further the clarity of the manuscript and to improve greatly the biological insights of this study.

We appreciate the positive response of all the reviewers for our discoveries regarding the role of the interaction of ERG with HDAC3 containing repressor complex in its leukemogenic activity. During the year of revisions of our paper several studies were published enlarging the relevance of ERG to variety of leukemias. Indeed, many experiments that we added to our previous revision demonstrate that HDAC3 inhibition is relevant to many subtypes of AML and ALL that are ERG dependent and, possibly, also to ERG driven prostate cancer. A recent paper by Dr. Brian Huntley's group (<https://www.ncbi.nlm.nih.gov/pubmed/36577137>), just published in *Blood*, suggests that the block of hematopoietic differentiation, a fundamental process in acute leukemias, may be caused by interaction of transcription factors with repressive complexes, as we show for ERG in our current manuscript. We have now cited this new paper in our discussion.

Regarding the concern of reviewer 1 about specificity control, we aimed to verify that the ERG-NCOR interaction is functionally specific to ERG-dependent AML by performing a co-immunoprecipitation assay also in ERG independent AML cell lines. We used an isotype control antibody (IgG raised in rabbit) as negative control. We found no detectable interaction between ERG and the NCOR2 complex in ERG-independent AML cell lines, supporting the hypothesis that this interaction is indeed limited to ERG-dependent AML cells (figure Rev1). Moreover, NCOR2 did not appear to be associated with the isotype control in either ERG-dependent or independent cells, indicating that the interaction with ERG is specific (figure Rev1).

Regarding the antibodies used in the assays, their catalog numbers are as follows: ERG (sc-271048), NCOR2 (sc-32298), and anti-HDAC3 (sc-11795). This information has now been added to the Materials and Methods section.

With regard to the physical characteristics of the ERG-NCOR-HDAC3 interaction, we would like to thank the reviewer for this comment. This issue has now been addressed in a new paragraph in the discussion section. We point out that the physical characteristics of the ERG-NCOR-HDAC3 interaction have yet to be fully elucidated. As yet, it is unknown whether the interaction is direct or indirect, and whether additional binding proteins may be involved. However, the purpose of our study was to determine the functional aspects of this interaction. In particular, we investigated its role in the development and maintenance of ERG-dependent leukemia and explored its potential as a therapeutic target. Hopefully, the revised version of our paper address all concerns raised and will be finally published.

A

B

Figure Rev1. Co-immunoprecipitation (Co-IP) assays in ERG-independent AML cell lines. Co-IP was performed using an antibody against endogenous ERG raised in rabbit, followed by blotting for NCoR2. We used an isotype control antibody (IgG raised in rabbit) as negative control. Representative figures of three independent experiments are shown. For a comprehensive description of the methods used, please refer to the manuscript. To illustrate that the low levels of endogenous ERG expression in THP1 and AML3 cells were not a result of insufficient total cell lysate loading, β -actin was employed as a loading control in the corresponding blots.

A. Co-IP in ERG-dependent cells (SKNO1) is shown as a representative positive control. This image was taken from the manuscript (Figure 8A). **B.** Co-IP for ERG and NCoR2 in THP1 cells (human acute monocytic leukemia cell line, ERG-independent) and in AML3 cells (human acute myelomonocytic leukemia, ERG-independent).